# Walking *Drosophila* navigate complex plumes using stochastic decisions biased by the timing of odor encounters

**Mahmut Demir[1†], Nirag Kadakia[1,2†], Hope D Anderson[1], Damon A Clark[1,3,4], Thierry Emonet[1,3,4]***

[1]Department of Molecular, Cellular and Developmental Biology, Yale University, New Haven, United States; [2]Swartz Foundation Fellow, Yale University, New Haven, United States; [3]Interdepartmental Neuroscience Program, Yale University, New Haven, United States; [4]Department of Physics, Yale University, New Haven, United States

**Abstract** How insects navigate complex odor plumes, where the location and timing of odor packets are uncertain, remains unclear. Here we imaged complex odor plumes simultaneously with freely-walking flies, quantifying how behavior is shaped by encounters with individual odor packets. We found that navigation was stochastic and did not rely on the continuous modulation of speed or orientation. Instead, flies turned stochastically with stereotyped saccades, whose direction was biased upwind by the timing of prior odor encounters, while the magnitude and rate of saccades remained constant. Further, flies used the timing of odor encounters to modulate the transition rates between walks and stops. In more regular environments, flies continuously modulate speed and orientation, even though encounters can still occur randomly due to animal motion. We find that in less predictable environments, where encounters are random in both space and time, walking flies navigate with random walks biased by encounter timing.

**\*For correspondence:**
thierry.emonet@yale.edu

[†]These authors contributed equally to this work

**Competing interests:** The authors declare that no competing interests exist.

## Introduction

Olfactory search strategies depend on both an animal's locomotive repertoire and the odor landscape it navigates. Navigational strategies have been investigated in a variety of odor plumes, each exhibiting a particular structure in space and time. The statistics of these plumes govern what information is available to the animal as it navigates, which in turn dictates the sequence of behaviors it can use to find its target. In some environments, such as the diffusion-dominated odor landscapes of *Drosophila* larvae, concentrations vary relatively smoothly from point to point. Accordingly, larvae can progress toward odor sources by sampling odor gradients spatially and temporally (*Gomez-Marin et al., 2011*; *Gepner et al., 2015*; *Hernandez-Nunez et al., 2015*; *Schulze et al., 2015*). Similarly, adult flies in gradients can walk up the gradient by monitoring the odor intensity difference across their antennae pairs (*Borst and Heisenberg, 1982*; *Gaudry et al., 2013*).

In the absence of stable gradients, odor landscapes may still be relatively simple when the airflow is laminar. This is true for modest wind speeds and in the near-surface laminar sublayer of turbulent flows, provided the source and average wind directions are not shifting and the surface is smooth (*Crimaldi and Koseff, 2001*). Variations in odor concentration are generally slow – odor encounters can last longer than seconds (*Crimaldi and Koseff, 2001*; *Álvarez-Salvado et al., 2018*). Furthermore, the largely unidirectional and steady wind provides a reliable cue about odor source location. Experiments in walking flies suggest that, in this case, high-frequency fluctuations in odor concentration might be ignored and upwind progress may result from temporal integration of the odor concentration (*Álvarez-Salvado et al., 2018*). Flies turn upwind at the onset of spatially uniform blocks

**eLife digest** When walking along a city street, you might encounter a range of scents and odors, from the smells of coffee and food to those of exhaust fumes and garbage. The odors are swept to your nose by air currents that move scents in two different ways. They carry them downwind in a process called advection, but they also mix them chaotically with clean air in a process called turbulence. What results is an odor plume: a complex ever-changing structure resembling the smoke rising from a chimney.

Within a plume, areas of highly concentrated odor particles break up into smaller parcels as they travel further from the odor source. This means that the concentration of the odor does not vary along a smooth gradient. Instead, the odor arrives in brief and unpredictable bursts. Despite this complexity, insects are able to use odor plumes with remarkable ease to navigate towards food sources. But how do they do this?

Answering this question has proved challenging because odor plumes are usually invisible. Over the years, scientists have come up with a number of creative solutions to this problem, including releasing soap bubbles together with odors, or using wind tunnels to generate simpler, straight plumes in known locations. These approaches have shown that when insects encounter an odor, they surge upwind towards its source. When they lose track of the odor, they cast themselves crosswind in an effort to regain contact. But this does not explain how insects are able to navigate irregular odor plumes, in which both the timing and location of the odor bursts are unpredictable.

Demir, Kadakia et al. have now bridged this gap by showing how fruit flies are attracted to smoke, an odorant that is also visible. By injecting irregular smoke plumes into a custom-built wind tunnel, and then imaging flies as they walked through it, Demir, Kadakia et al. showed that flies make random halts when navigating the plume. Each time they stop, they use the timing of the odor bursts reaching them to decide when to start moving again. Rather than turning every time they detect an odor, flies initiate turns at random times. When several odor bursts arrive in a short time, the flies tend to orient these turns upwind rather than downwind.

Flies therefore rely on a different strategy to navigate irregular odor plumes than the 'surge and cast' method they use for regular odor streams. Successful navigation through complex irregular plumes involves a degree of random behavior. This helps the flies gather information about an unpredictable environment as they search for the source of the odor. These findings may help to understand how other insects use odor to navigate in the real world, for example, how mosquitoes track down human hosts.

of natural (*Álvarez-Salvado et al., 2018*; *Steck et al., 2012*) or optogenetic fictive odors (*Bell and Wilson, 2016*), and turn downwind or initiate a local search when the odor is lost.

Steady odor ribbons have provided an informative experimental paradigm used extensively in insect navigation studies (*Budick and Dickinson, 2006*; *van Breugel and Dickinson, 2014*). Like near-surface flows and spatially uniform odor blocks, ribbons are spatiotemporally simple. Walking and flying moths accelerate and turn upwind upon entering the straight ribbons, and cast perpendicular to the wind or counterturn when losing it (*Mafra-Neto and Cardé, 1994*; *Cardé and Willis, 2008*; *Kuenen and Carde, 1994*; *Kanzaki and Sugi, 1992*; *Vickers and Baker, 1994*; *Kennedy and Marsh, 1974*; *Baker and Haynes, 1989*; *Haynes and Baker, 1989*; *Baker and Vickers, 1997*). Flying flies navigate them similarly, combining upwind surges with counterturns back into the ribbon after passing through. In these plumes, odor encounters are very brief (~100 ms) and also exhibit some degree of temporal irregularity due to the animal's self-motion as it randomly crosses the ribbon (*van Breugel and Dickinson, 2014*). Still, the locations of encounters are highly predictable, and this spatial regularity is naturally exploited by reflex-dominated strategies such as surging and counterturning (*Pang et al., 2018*).

By contrast, in the bulk of turbulent flows (*Cardé and Willis, 2008*; *Murlis et al., 1992*; *Murlis et al., 2000*; *Riffell et al., 2008*; *Yee et al., 1993*) or on rough surfaces where shifting winds and obstacles such as grass, shrubs, and branches can perturb the laminar boundary layer (*Schlichting, 1960*; *Hunt et al., 1978*), odor landscapes are irregular in both space and time. Measurements of odor concentrations in forests (*Murlis et al., 2000*; *Webster and Weissburg, 2001*; *Moore et al.,*

*1994*) show that not only are local concentration gradients less indicative of odor source location, but importantly, odor encounters are intermittent, occurring as a random sequence of brief bursts. Theory suggests that in complex intermittent plumes, the timing of odor encounters may provide important information to the navigator (*Balkovsky and Shraiman, 2002*; *Vergassola et al., 2007*). Indeed, moths follow tight trajectories upwind while navigating within a turbulent plume, much narrower than those in steady ribbons (*Mafra-Neto and Cardé, 1994*; *Kanzaki and Sugi, 1992*; *Baker and Haynes, 1989*). These narrow tracks were recapitulated for moths navigating pulsed ribbons, provided the pulse frequency was high enough, again implicating encounter timing in upwind progress (*Mafra-Neto and Cardé, 1994*).

There is an important distinction between experiments informed by steady odor ribbons versus those informed by spatiotemporally complex plumes. In steady ribbons, navigational behaviors can be tied to individual odor encounters because the location of the ribbon can be measured and the time-dependent odor signal perceived by the animal can be inferred from its trajectory. In spatio-temporally complex plumes, behaviors can at most be correlated with plume statistics, as the time when each individual filament hits the animal is unknown. In this context, analyzing how animals use the timing of individual encounters to navigate would require simultaneous measurement of behavior with odor.

Here we identified the navigational principles of walking fruit flies in spatiotemporally complex odor plumes resembling those in naturalistic settings. In particular, we investigated how these navigational principles are shaped by the temporal features of the individual odor encounters made with the rapidly fluctuating plume. We exploit a technical advance that relaxes the tradeoff between restricting odor dynamics and animal motion: an attractive odor that can be imaged in real time with unrestrained walking flies. By passing this odor in a laminar airflow and perturbing it with random lateral air jets, we generate a spatiotemporally complex plume whose statistics approximate those of turbulent plumes near boundaries (*Celani et al., 2014*; detailed comparison with theory in Results). This odor allows us to study walking fly olfactory navigation by directly connecting navigational behaviors to individual odor encounters.

Consistent with prior studies (*Budick et al., 2007*; *David et al., 1982*), we find that flies on average walk upwind within the odor plume cone. However, upwind bias does not result from an accumulation of orientation changes following every odor encounter. Instead, flies execute stochastic, stereotyped 30-degree saccades at a rate independent of the duration or frequency of odor encounters. Upwind bias results not from modulating turn magnitude or frequency but rather turn direction: the randomly-occurring saccades are more likely to be oriented upwind when the frequency of odor encounters – but not their duration or concentration – is high, suggesting an important role for precise odor timing detection (*Gorur-Shandilya et al., 2017*; *Szyszka et al., 2014*; *Shusterman et al., 2011*; *Park et al., 2016*). Prior studies have shown that flies increase the walking speed at the onset of uniform odor blocks (*Álvarez-Salvado et al., 2018*; *Jung et al., 2015*; *Gao et al., 2013*). In our spatiotemporal plume, flies spend only a fraction of time (~15%) experiencing detectable odor concentrations, and we expectedly do not find an appreciable increase in walking speed. However, flies do markedly modulate their rate of walking and stopping. In contrast to turn decisions, the rates of these walk-stop transitions are strongly tied to the frequency of encounters. We model stops and walks as a double, inhomogeneous Poisson process and find using maximum likelihood estimation and cross-validation that stop rates reset at every encounter before decaying back to a baseline rate. This suggests that individual encounters prolong the flies' tendency to continue walking but only for a brief time. Meanwhile, walks are triggered by accumulating evidence from multiple encounters while stopped. Using agent-based simulations, we show that this modulation of stops and walks shaped by the timing of odor encounters greatly enhances navigation performance. Together, our results suggest that navigation within spatiotemporally complex odor plumes is shaped by the sequence of encounters with individual odor packets. Both electrophysiological and behavioral measurements indicate that *Drosophila* – along with other insects, mammals, and crustaceans, among others – can precisely encode odor timing within their signal transduction cascade (*Gorur-Shandilya et al., 2017*; *Park et al., 2016*; *Smear et al., 2011*; *Schaefer and Margrie, 2007*). Our findings suggest that *Drosophila* leverage this capability to navigate their olfactory world.

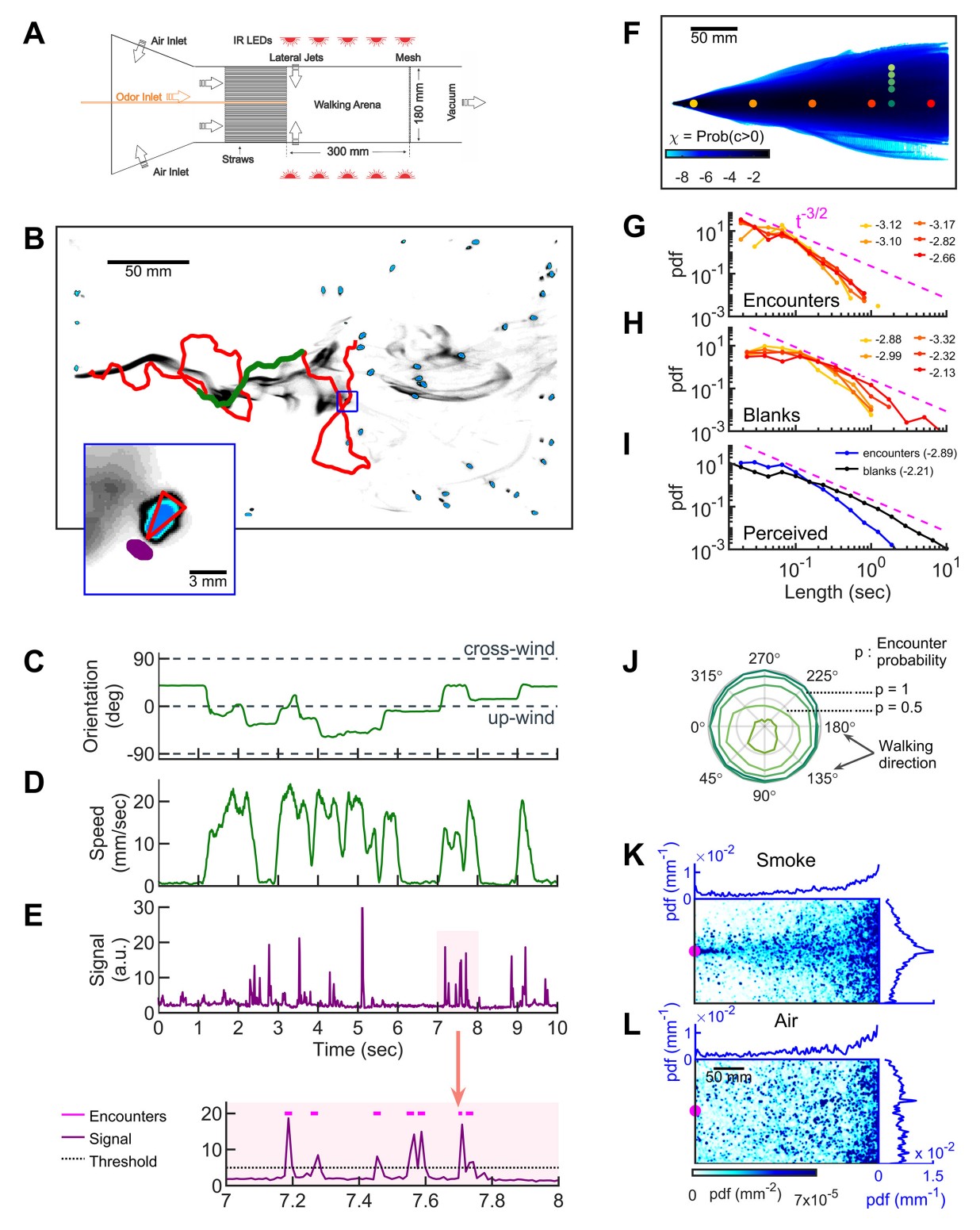

**Figure 1.** Simultaneous visualization of odor and fly behavior. (**A**) Experimental apparatus. The main flow (150 mm/s) is perturbed by lateral jets (1500 mm/s) that alternate stochastically with a characteristic time scale of 100 ms. (**B**) Walking arena with flies (blue), odor intensity (gray), and a representative trajectory of a navigating fly (red). Green: the portion of the trajectory plotted in C-E. Note that since the odor environment is fluctuating, the image in B only represents the environment at a random time point. Blue rectangle: the area shown in the inset. Inset: blue: fly; gray: odor intensity; red triangle: orientation of the fly; purple: the virtual antenna in which the odor intensity is averaged as a proxy for the signal perceived

*Figure 1 continued on next page*

*Figure 1 continued*

by the fly. Simultaneous orientation (C), speed (D), and perceived stimulus (E) of the fly while it is navigating in the intermittent plume (green portion of the trajectory in B). Dashed lines in C indicate up-wind and cross-wind orientations. Orientation and speed were smoothed with a 100 ms sliding box filter. The shaded area in E (top) is plotted at a larger scale (bottom) with the sensory threshold (dotted line) used to identify the odor encounters (magenta lines above the signal trace in purple; see also *Figure 1—figure supplement 2* and Materials and methods). (F) The fraction of time that odor concentration is above the sensory threshold (i.e. intermittency) at fixed locations. Image intensity is median-filtered (square filter size 2.3 mm), and the likelihood that the intensity is above the sensory threshold, averaged over all frames of the video. Red dots: positions of encounter and blank duration distributions plotted in G-H. Green dots: positions of encounter probability distributions plotted in J. (G-H) Distributions of encounter and blank durations, respectively, at positions color-coded in F. The pink dash line shows the $t^{-3/2}$ expected from theory for turbulent flow (*Celani et al., 2014*). Exponents of the power law fit to the tail of the distributions are indicated with the same color code as in F. (I) Odor encounter and blank durations perceived by navigating flies. Values in the parenthesis indicate the exponent of the power law fit to the tail of the distribution. (J) Probability (r axis) to have an odor encounter within 1 s while walking with a speed of 10 mm/s, starting from positions color-coded in F, as a function of walking direction (theta axis). (K) Probability distribution functions (pdf) of fly positions in the arena for the complex smoke plume as in A (n = 1073 trajectories). (L) Same without smoke but with the same complex wind pattern as in K (n = 502). Magenta: location of the source. Blue curves: marginal pdfs.

The online version of this article includes the following figure supplement(s) for figure 1:

**Figure supplement 1.** Attraction to smoke is olfactory and dose-dependent, closely mimicking to ethyl acetate and apple cider vinegar.
**Figure supplement 2.** The average turning position of flies in straight plume agrees with smoke intensity.
**Figure supplement 3.** Intermittency in the complex plume.

## Results

### Visualizing dynamic odor plumes simultaneously with fly behavior

To investigate how freely-walking insects navigate odor plumes that are complex in both space and time, we developed a wind-tunnel walking assay for *Drosophila melanogaster* (*Figure 1A*). The large size of our 2D arena (300 × 180 × 1 cm³) allowed us to simultaneously image several flies in the dark with minimal mutual interactions. The main flow was set to 150 mm/s, chosen as sufficiently strong for flies to tax upwind, but not so strong that they remained stationary (*Yorozu et al., 2009*). Plumes that fluctuated in space and time were generated by injecting odors at the center of an air comb and perturbing the laminar flow with lateral jets stochastically alternating at a Poisson rate of 10/s. To visualize the flow, we injected smoke, which is turbid, into the center of the air comb and imaged it in the infrared at 90 Hz. Serendipitously, we noticed that when we placed starved flies in the assay with the fluctuating smoke, flies walked upwind toward the source (*Figure 1B* , *Video 1*) in a manner reminiscent of their behavior when we injected an attractive odor such as ethyl acetate. We reasoned that if this attraction to smoke were olfactory, the imaged smoke intensity could then provide a proxy for odor concentration, allowing us to visualize dynamic odor plumes simultaneously with fly behavior (*Figure 1C–E* and *Video 2*).

Smoke is a complex stimulus (*Figure 1—figure supplement 1A*), containing not only $CO_2$ and volatile chemicals, but also heat, humidity, and airborne particles. We, therefore, set out to verify that the attraction to smoke is olfactory. For this purpose, we used a simplified environment consisting of a standing odor ribbon, which we generated in our assay by maintaining the laminar flow and odor injection, but turning off the lateral jets (*Figure 1—figure supplement 1B*). First, we compared behavioral statistics in smoke to those in the attractive odors ethyl acetate (EA) and apple cider vinegar (ACV). The likelihood that flies were in the narrow band near the smoke ribbon increased with smoke concentration (*Figure 1—figure supplement 1C*), before saturating at a sufficient dose (*Figure 1—figure supplement 1D*), a result reproduced in both EA and ACV (*Figure 1—figure supplement 1E–H*). We then tested contributions from carbon dioxide sensing and vision, using Gr63a[-/-] (*Jones et al., 2007*) and norpA[-/-] (*Bloomquist et al., 1988*) mutants, respectively. Both mutants retained the ability to localize the odor source at a level comparable to wild-type flies (*Figure 1—figure supplement 1I–J*). To test whether humidity played a major role, we saturated the airflow with 80% humidity and found that source localization was reduced but still significantly above random (*Figure 1—figure supplement 1I–J*). Finally, we tested the olfaction directly using Orco[-/-] mutants (*Larsson et al., 2004*), as well as anosmic flies (Gr63a[-/-], Orco[-/-], Ir8a[-/-], and Ir25a[-/-]; *Ramdya et al., 2015*). In both sets of mutants, the ability to find the odor source was completely abolished. Orco[-/-] mutants (but not the anosmic flies) exhibited a slight repulsion to the smoke ribbon, which we attributed to an aversive response to carbon dioxide (*Figure 1—figure supplement*

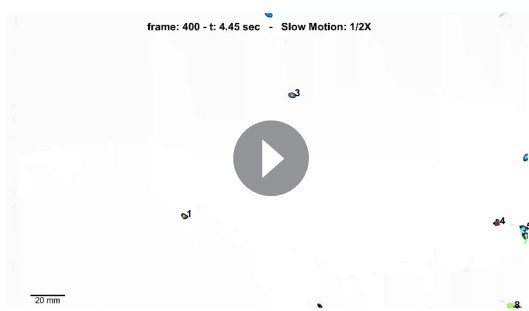

**Video 1.** Starved wild-type (CS) female flies navigating in the intermittent smoke plume.
https://elifesciences.org/articles/57524#video1

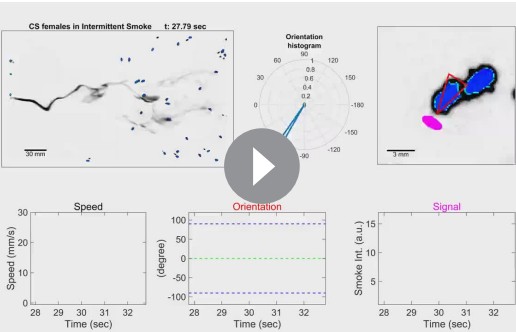

**Video 2.** Simultaneous quantification of odor stimulus and fly behavior. Starved wild-type (CS) female flies navigating in the intermittent smoke plume.
The trajectory of a single fly (same as in *Figure 1A*) is displayed with its speed, orientation, and perceived signal. Green and blue dashed lines in the orientation panel represent up-wind and cross-wind orientations, respectively.
https://elifesciences.org/articles/57524#video2

*1I-J*; *Larsson et al., 2004*; *Suh et al., 2004*). Thus, flies' attraction to smoke is driven mainly by olfaction.

To quantify the time-dependent stimulus experienced by each fly during navigation, we averaged the signal intensity in a small area (1.10 mm$^2$) near its antennae (*Figure 1B* inset, *Figure 1E*, and *Video 2*). The onset and offset of odor encounters were defined as the times when the signal crossed a sensory threshold, which we set to 2.5 SD ($\sigma$) above the background noise. We refer to the periods when the odor is above threshold as 'odor encounters,' or 'encounters' for short, and periods when the odor is below the threshold as 'blanks.' We verified that the results and conclusions presented below remained unchanged for thresholds between 2.0$\sigma$ and 3.5$\sigma$. Using the 2.5$\sigma$ threshold, the error in the timing of odor encounters was estimated to be less than 25 ms (*Figure 1—figure supplement 2* and Materials and methods). Using this setup, we then examined how walking flies navigate odor plumes that fluctuate in both space and time.

## Within the odor plume, odor encounters are brief, frequent, and unpredictable

We first quantified the statistics of the odor environment the flies must navigate. Our odor plume was highly intermittent, composed of spatiotemporally localized filaments breaking continuously in time (*Figure 1B* and *Video 1*, lateral jets on). Odor mean intermittency – the fraction of time the odor was above the sensory threshold – ranged several orders of magnitude in the conical extent of

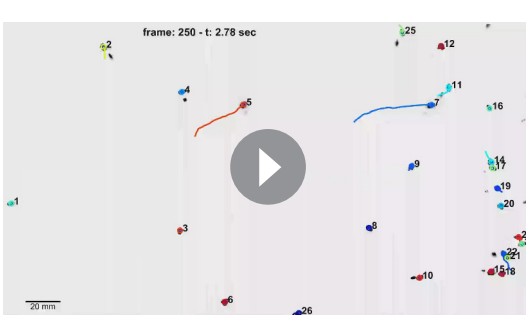

**Video 3.** Starved wild-type (CS) female flies navigating in the intermittent smoke plume. The smoke valve is turned on/off at 15 s blocks while the wind is kept fluctuating.
https://elifesciences.org/articles/57524#video3

the plume (*Figure 1F*). The average signal intermittency was low across the arena, increasing from about 10$^{-6}$ at the border of the plume to about 0.12–0.39 at the center line, depending on the distance to the source (see also *Figure 1—figure supplement 3A*). Still, navigating flies perceived intermittencies ranging over a decade and a half, with an average intermittency around 0.11 (*Figure 1—figure supplement 3B*), resembling values measured (*Murlis et al., 2000*) in natural settings. At fixed locations from the source, odor encounter (*Figure 1G*) and blank (*Figure 1H*) durations spanned a wide range of time scales (exponents ranging from −3.3 close to the source to −2.1 far from the source), gradually approaching the power law $\sim t^{-3/2}$ theoretically predicted for turbulent odor plumes in the atmospheric

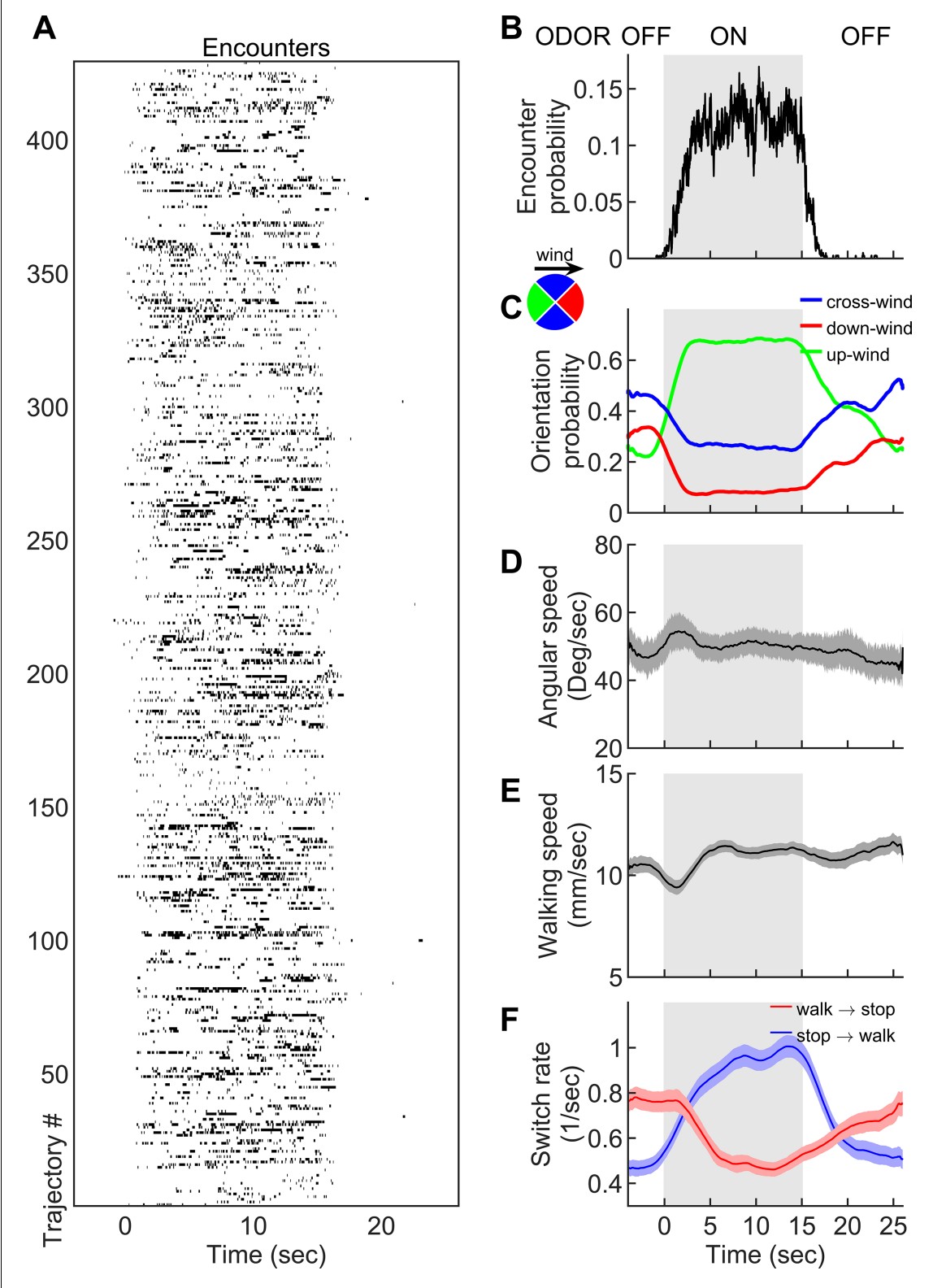

**Figure 2.** Navigation within intermittent complex odor plumes comprises walk-stop transitions and upwind orientation. (**A**) The odor encounters experienced by flies navigating the intermittent smoke plume as in *Figure 1B* (see also *Video 3*). While the wind is continuously perturbed by the lateral jets, the odor is turned on/off in 15 s blocks (gray shading). Rows indicate independent trajectories (*n* = 429) obtained from 267 flies. (**B-F**) Quantities averaged over all trajectories in A as a function of time. (**B**) Probability of having an encounter. (**C**) Probability of being in up-wind (green), *Figure 2 continued on next page*

eLife Research article

Neuroscience | Physics of Living Systems

*Figure 2 continued*

cross-wind (blue), and down-wind (red) orientations estimated in 90-degree quadrants as shown in the circle with the same color codes. C-E include only time points for which the fly was walking (*v* > 2 mm/s). (D) Angular speed. (E) Walking speed. (F) Walk-to-stop (red) and stop-to-walk (blue) switching rates. All quantities in B-F are smoothed with a 5 s sliding box filter. Error bars indicate SEM.

The online version of this article includes the following figure supplement(s) for figure 2:

**Figure supplement 1.** Smoke elicits behavior similar to natural odorants.

boundary layer (*Celani et al., 2014*). The distribution of encounter and blank durations experienced by navigating flies spanned an even greater range and were closer to a power law (*Figure 1I*). On average, flies experienced brief odor encounters (mean duration ~200 ms) at a mean frequency of 4 Hz. Even beyond their variability and brevity, encounters were also highly unpredictable in location. To quantify this, we calculated the likelihood to receive an odor encounter in 1 s, assuming one walks straight at 10 mm/s radially outward from a fixed point. Predictability in the location of future odor encounters would then manifest as a directional dependence of this likelihood. Within the conical extent of the plume, the likelihood was nearly isotropic with respect to walking direction, whereas near the plume edges, likelihoods were skewed toward the centerline of the plume cone (*Figure 1J*). Within the conical extent of the odor plume, therefore, the location of future odor encounters was uncertain. Despite this uncertainty, flies remained largely in the plume cone and were able to successfully locate the odor source (*Figure 1K*). However, during fluctuating winds without odor, they could not locate the source (*Figure 1L*).

## Stopping and turning comprise the bulk of the navigational repertoire within a spatiotemporally complex odor plume

How are fly orientation and speed shaped on average by an odor signal exhibiting this degree of spatiotemporal complexity? To compare these behaviors to those in an odorless environment, we presented the complex plume in 15 s blocks by closing and opening the odor valve every 15 s but maintaining the alternating lateral jets throughout the trial. This produced an environment in which a 15 s block of complex odor plume alternated with a 15 s block of fluctuating wind only (*Video 3*). When the odor was on, odor encounters were frequent, but randomly experienced in time (*Figure 2A–B*). As expected, flies were more likely to be oriented upwind when the odor was on (*Figure 2C*), as previously reported (*Álvarez-Salvado et al., 2018*; *Steck et al., 2012*; *Bell and Wilson, 2016*; *Budick and Dickinson, 2006*; *Kennedy and Marsh, 1974*; *Murlis et al., 1992*; *Flügge, 1934*). However, unlike for flies walking into a spatially homogeneous odor block (*Álvarez-Salvado et al., 2018*; *Jung et al., 2015*; *Gao et al., 2013*), changes in average angular speed were minor, with a less than 10% change between blocks (*Figure 2D*). Walking speeds were similarly unmodulated, again in contrast to walking flies in homogenous odor blocks (*Figure 2E*; *Álvarez-Salvado et al., 2018*). This is not inconsistent, however, since encounters were so brief (~200 ms), the integration timescales for speed modulation measured previously (*Álvarez-Salvado et al., 2018*) would only produce <10% increase in either ground or angular speed.

Though changes in ground speed were minor, we noticed a high incidence of stopping in our spatiotemporally complex plume (*Figure 1D*). The prevalence of immobility has been noted before in walking flies navigating homogenous odor blocks (*Álvarez-Salvado et al., 2018*), though its role in navigation was not investigated. We suspected that stopping might form a critical component of intermittent plume navigation for walking flies. Indeed, walk-to-stop and stop-to-walk transition rates were strongly modulated during the transitions between odorized and non-odorized blocks (*Figure 2F*). Natural odors ACV and EA elicited similar navigational trends in angular and ground speeds, orientation, and stopping rate when presented in these 15 s blocks (*Figure 2—figure supplement 1*). Together, this suggested that turning and stopping comprised the bulk of the navigational repertoire for walking flies in spatiotemporally complex plumes. This prompted us to next examine how the sequence of individual odor encounters experienced by navigating flies precisely shapes their decisions to turn, walk, and stop.

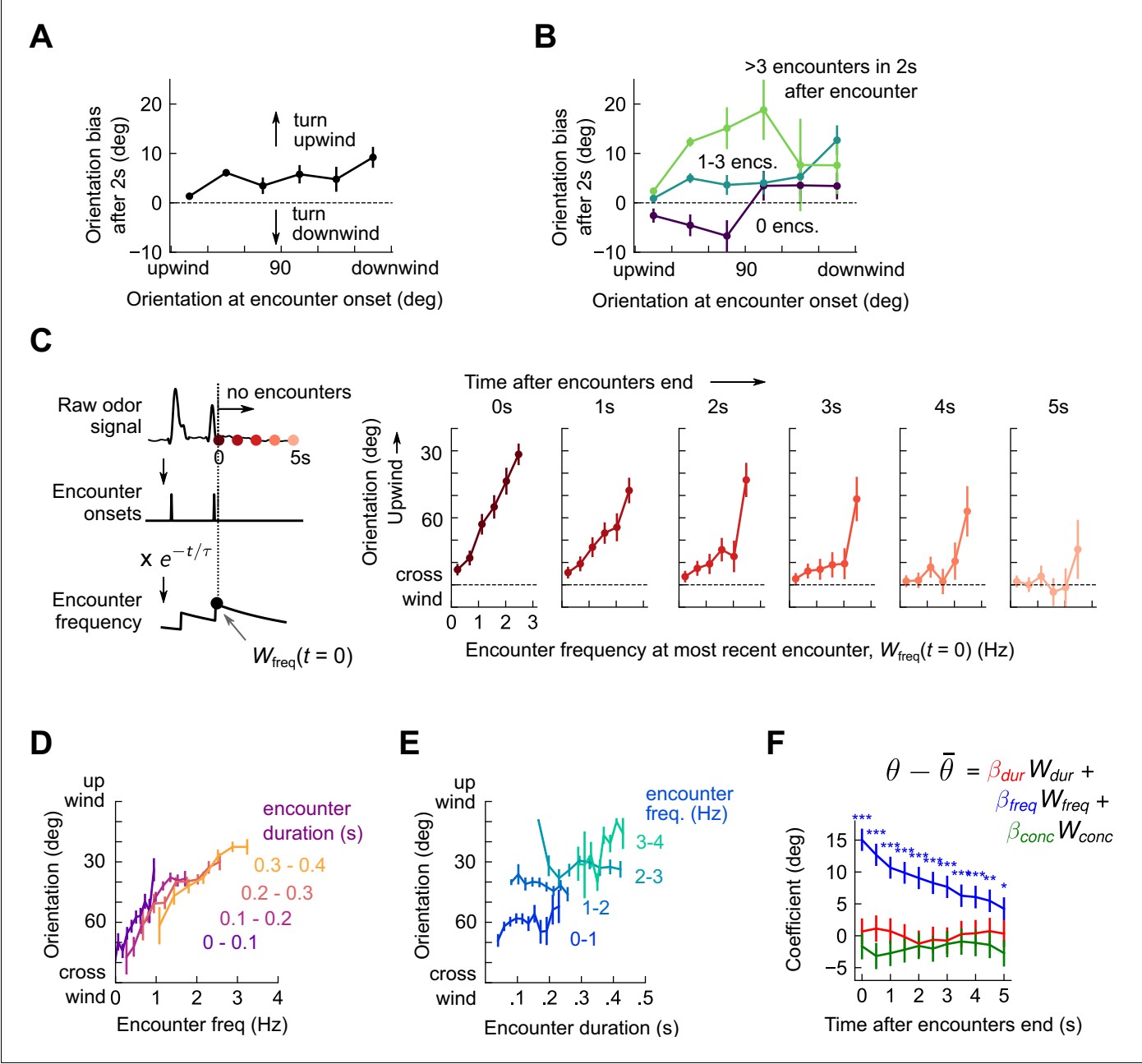

**Figure 3.** Flies use encounter frequency to bias orientation upwind. (A) Orientation 2 s after an encounter, as a function of orientation at encounter onset ($n$=5040). The mean orientation change for random times is subtracted out. (B) Same data, now binned by the number of subsequent encounters in the 2 s window. (C) Orientation as a function of encounter frequency at the most recent encounter, $W_{freq}(t = 0)$, for various times $t > 0$ after encounters have been interrupted. Encounter frequency is defined by convolving the binary vector of encounter onsets with an exponential filter of timescale $\tau$ = 2 s. $t = 0$ defines the time of the most recent encounter, and the individual plots show fly orientation as a function of the encounter frequency at this time, $W_{freq}(t = 0)$. Orientation biases strongly upwind with frequency and this correlation vanishes after ~5 s. (D) Orientation versus encounter frequency when encounter duration is held fixed within a small range, for various ranges. (E) Orientation versus encounter duration when encounter frequency held fixed within a small range. (F) Estimated regression coefficients for a trilinear fit of fly orientation to encounter frequency, encounter duration, and signal intensity. Each of the independent variables has been standardized. Coefficients are plotted for various times after encounters are interrupted (as in C). Statistical significances using a 2-tailed t-test are shown next to curves; if no stars are shown, the coefficients are not statistically distinct from 0. The data indicate that orientation correlated with encounter frequency but not encounter duration or signal intensity.

The online version of this article includes the following figure supplement(s) for figure 3:

**Figure supplement 1.** Encounter-elicited orientation change in time.

## Upwind orientation results from repeated odor encounters, not cumulative odor exposure time

Flies reorient upwind soon after flying into an odor ribbon (*van Breugel and Dickinson, 2014*) or walking into a homogeneous odor block (*Álvarez-Salvado et al., 2018*). We therefore calculated the change in fly orientation following an individual encounter, finding that within 2 s of an encounter onset, flies of any orientation biased their orientation upwind (*Figure 3A* and *Figure 3—figure supplement 1*). Since encounter frequency was on the order of a few Hz, flies receiving one encounter were likely to receive more within the 2 s window. Upwind bias may therefore reflect an accumulated effect from repeated odor encounters. Partitioning the data into encounters followed by 0, 1-3, or 4 + further encounters within 2 s, we found that odor encounters followed closely by many others elicited much stronger upwind bias than did isolated ones (*Figure 3B*). To quantify this more precisely, we calculated a running average of encounter frequency $W_{freq}(t)$ by convolving the binary vector of encounter onset times with an exponential filter (timescale $\tau$ = 2 s), and plotted upwind orientation as a function of encounter frequency (*Figure 3C*). All orientations were reflected over the *x*-axis, whereby 0° is upwind and 180° is downwind. The trend was strongly monotonic, with an intercept of 88.6° at 0 Hz – flies experiencing no encounters were oriented nearly equally upwind and downwind – and a slope of 21.6°/Hz ($p < 10^{-4}$) – flies experiencing a frequency of 3 Hz would be oriented just 25° off the upwind direction. If no further encounters were received, this monotonic trend dropped steadily to a slope of 4.5°/Hz (not significantly different from 0, $p > 0.05$) after 5 s (*Figure 3C*). This suggests that repeated interactions with the plume biased the fly upwind, and after some time without encounters, flies were again uniformly oriented.

The amount of time a fly is exposed to odor increases with each subsequent encounter. Does upwind bias result from the number of individual odor interactions, the cumulative duration of these encounters, or both? If, for example, all encounters were 200 ms long, then tripling encounter frequency would also triple perceived odor duration – frequency and duration would be perfectly correlated. But if orientation depended on odor duration alone, the dependency on frequency noted above would arise simply as a consequence of this correlation. Prior results suggest that walking flies bias orientation and speed by filtering odor in time (*Álvarez-Salvado et al., 2018*), so we suspected that odor duration might contribute to some or all of the upwind bias. To investigate this possibility, we defined a running average of odor duration $W_{dur}(t)$ analogously to $W_{freq}(t)$ by exponentially filtering the binary vector of odor intermittency (1 during encounters, 0 during blanks). We disassociated $W_{freq}(t)$ and $W_{dur}(t)$ by holding one constant to a small range, and plotting upwind orientation against the other. Surprisingly, with this analysis, only the correlation of orientation with encounter frequency remained (*Figure 3D-E*). We also investigated the possibility that odor concentration contributed to upwind turning by defining $W_{conc}(t)$ analogously using the raw signal. While we have not quantified the exact relationship between odor concentration and image intensity, our dose-response results (*Figure 1—figure supplement 1C-D*) suggest that they are monotonically related, so a correlation would exist to first order. Linearly regressing upwind orientation simultaneously against $W_{freq}(t)$, $W_{dur}(t)$, and $W_{conc}(t)$, revealed $W_{freq}(t)$ as the sole explanatory variable ($p < 1e-6$, $p > 0.05$, $p > 0.05$, respectively; *Figure 3F*). Together, these results indicate that in the intermittent, spatiotemporally complex plumes in this experiment, upwind orientation was driven by the frequency, but not by the duration or concentration, of odor encounters.

## Odor encounters bias turn direction but not turn likelihood or turn magnitude

The lack of a clear upwind bias following an isolated encounter (*Figure 3A* and *Figure 3—figure supplement 1B*) suggested that reorientations may not be simply an encounter-elicited reflex. To characterize reorientations, we first thresholded angular speed to identify turn events (*Figure 4—figure supplement 1*, and Materials and methods). We found that individual turns occurred not in a continuum of angles but rather in discrete saccades of 30°±10° either left or right (*Figure 4A*), consistent with previous studies in non-odorized environments (*Geurten et al., 2014*). Moreover, the contribution to upwind bias from the inter-saccade sections of the trajectories was not significant (*Figure 4B*). This indicates that the discrete saccadic turns were responsible for upwind progress during navigation. The waiting time between saccades obeyed an exponential distribution with timescale $\tau$ = 0.75 s±0.17, or a Poisson rate of about 1.3 turns per second (*Figure 4C*). Surprisingly, this

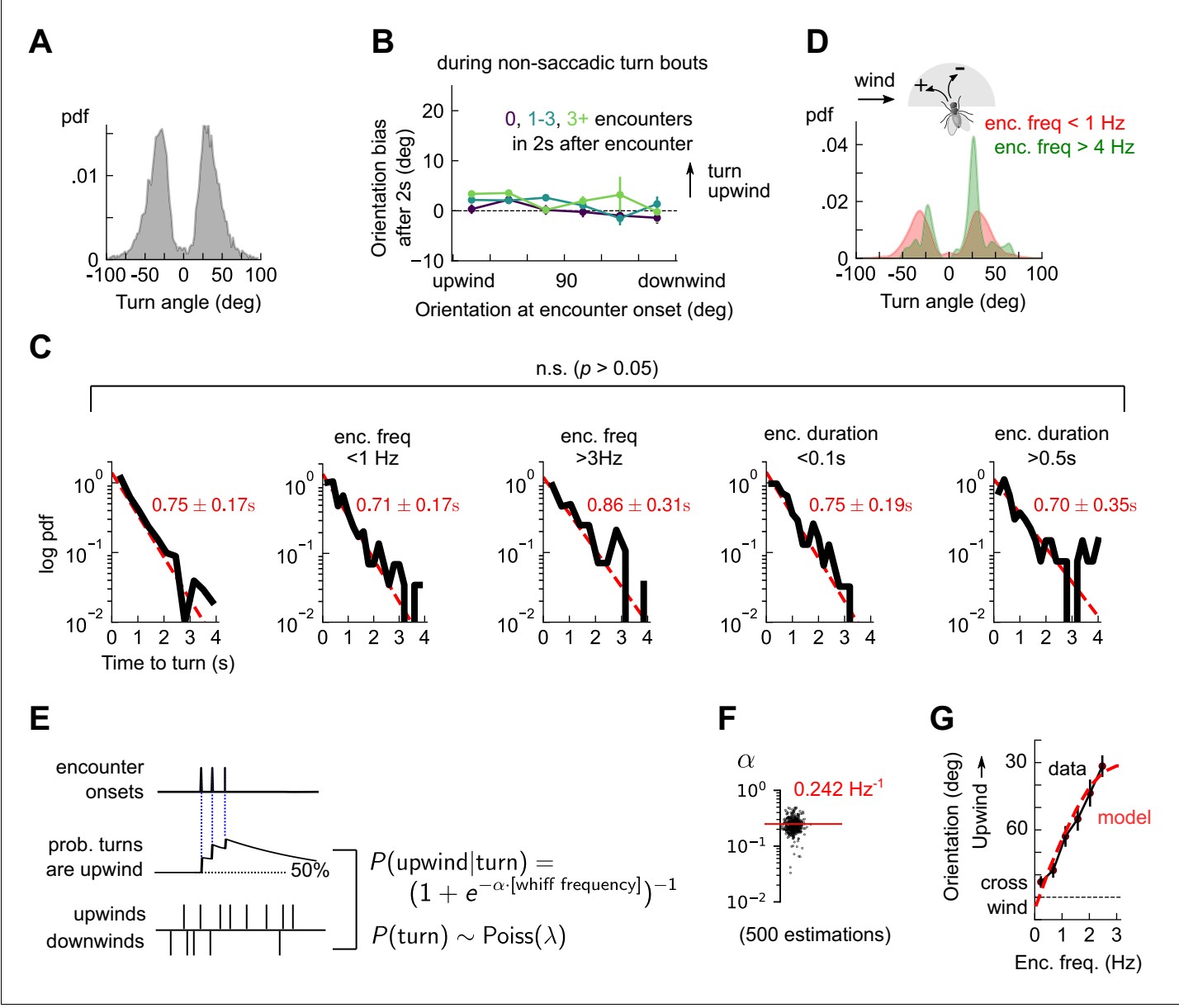

**Figure 4.** Odor encounters bias turn direction but not turn rate or turn magnitude. (A) Distribution of change in orientation following a turn. The discreteness of turn angle (two narrow peaks in pdf) was verified to be insensitive to the threshold used to determine turns (*Figure 4—figure supplement 1*). (B) Cumulative change in orientation over non-turning bouts ('straight' segments) 2 s after an encounter, versus orientation at encounter onset (compare with *Figure 3B*). Data are partitioned into encounters followed by 0, 1-3, or 4+ subsequent encounters in the following 2 s. (C) Leftmost plot: Distribution of time until a turn. Red line: maximum likelihood fit to an exponential distribution, with mean 0.75±0.17 s (distribution generated by bootstrapping). Remaining plots: Same, now for times at which encounter frequency is low (<1Hz; 2nd plot) or high (>3Hz; 3rd plot), or times at which encounter duration is low (<100 ms; 4th plot) or high (>500 ms; 5th plot). Fits are 0.71±0.17 s, 0.86±0.31 s, 0.75±0.19 s, 0.70±0.35 s, respectively, none of which are statistically distinct from the data for all turns ($p > 0.05$, 2-tailed t-test). (D) Distribution of turn angles during low (< 1 Hz) or high (> 4 Hz) encounter frequency bouts (compare A). (E) Model of fly turning. Turn events obey a Poisson process with timescale $\tau_T$=0.75s (C). Turn direction is chosen randomly at each turn time, where the probability $p_T$ that the turn is directed upwind is modeled by

$p_T = (1 + e^{-\alpha W_{freq}})^{-1}$ (Materials and methods). $p_T$ is therefore a sigmoidal function of the frequency of odor encounters $W_{freq}$, where the gain parameter $\alpha$ represents the steepness of the sigmoid. (F) Distribution of gain parameter $\alpha$, estimated for 500 distinct subsets of the data. The distribution is highly peaked, indicating its robustness. The median estimate of $\alpha$ is 0.242 Hz$^{-1}$. (G) Upwind orientation versus encounter frequency for data (black) and model prediction (red).

The online version of this article includes the following figure supplement(s) for figure 4:

**Figure supplement 1.** Quantification of turn detection.

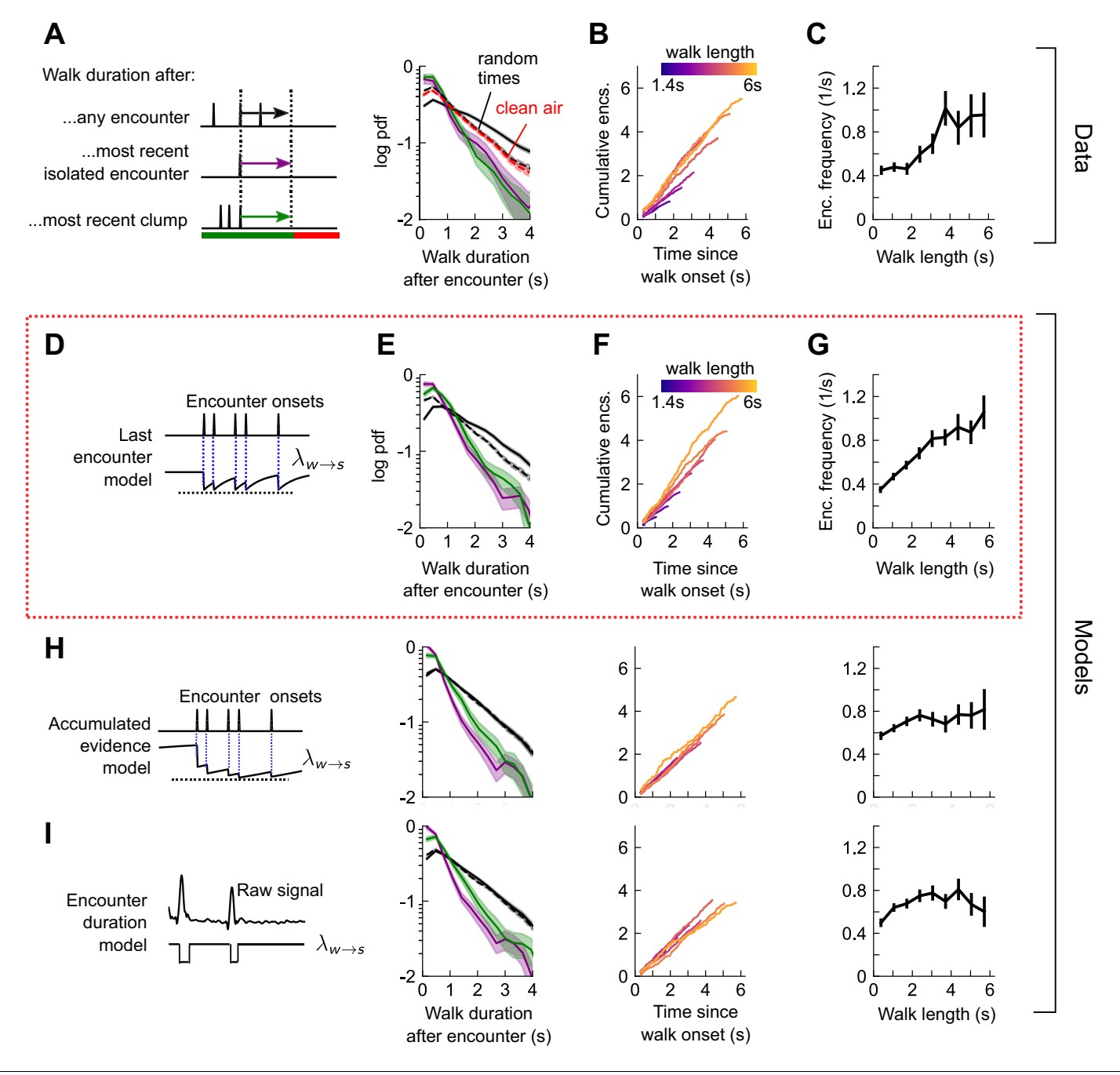

**Figure 5.** Stop decisions are stochastic events whose rate is modulated by the timing of the most recent encounter. (A) Distribution of walking duration following any odor encounter, or the most recent encounter or clump before a stop. Dashed: distribution for randomly chosen times. Red dashed: clean air control. (B) Cumulative encounter counts since walk onset, for various walk durations. (C) Average encounter frequency versus duration of walk bout. (D) In the last encounter model, stop decisions are modeled as a Poisson process with an inhomogeneous rate $\lambda_{w\to s}(t)$, where $\lambda_{w\to s}(t)$ resets to a fixed value at every encounter, then decays back to baseline. This is modeled by $\lambda_{w\to s}(t) = \lambda_0 + \Delta\lambda e^{-\Delta T(w(t))/\tau_s}$, where $\Delta T(w(t))$ is the time since the most recent encounter (Materials and methods for details). Median of estimated parameters are $\lambda_0 = 0.78\text{s}^{-1}$, $\Delta\lambda = -0.61\text{s}^{-1}$, $\tau_s = 0.25\text{s}$ (*Figure 5—figure supplement 2*). (E-G) Analogs of A-D using data generated by the model. (H) Analogs of E-G for the accumulated evidence model. In this model, $\lambda_{w\to s}(t)$ decreases at every encounter, but remains at a lower value when encounters are more closely spaced. We model this with

$$\lambda_{w\to s}(t) = \lambda_0 + \lambda_1 / \left(1 + \lambda_2 \int e^{\frac{t-t'}{\tau_s}} w(t')dt'\right).$$ (I) Analogs of E-G for the encounter duration model, in which $\lambda_{w\to s}(t)$ switches between a low value during encounters and a higher value during blanks.

*Figure 5 continued on next page*

*Figure 5 continued*

The online version of this article includes the following figure supplement(s) for figure 5:

**Figure supplement 1.** Stop and walk decisions depend on encounter timing.
**Figure supplement 2.** Distributions of estimated parameters for walk-to-stop models.

turn rate was insensitive to either encounter frequency or duration (*Figure 4C*). This presented a puzzle: if flies turned left and right at discrete angles and a constant rate, they were effectively executing a random walk on the circle. Since angular random walks randomize orientations in time, how would flies orient upwind? Partitioning the turn angle distribution into bouts of low (< 1 Hz) and high (> 4 Hz) encounter frequency resolved this puzzle. For high frequencies, the distribution of turn angles exhibited the same ±~30° peaks, but now with an upwind lobe much larger than the downwind one (*Figure 4D*). Thus, odor encounter frequency biased the direction of turns, while leaving the magnitude and rate of turns unchanged.

These findings could be recapitulated with a simple stochastic model of turning, in which walking flies execute Poisson turns at a constant rate. The magnitude of each turn is chosen randomly from the measured distribution, and the likelihood $p_T$ that the turn is directed upwind is a sigmoidal function of the encounter frequency. Specifically, $p_T = \left(1 + \exp\left(-\alpha W_{freq}\right)\right)^{-1}$ (*Figure 4E*). This model produces unbiased turns ($p = 0.5$) in the absence of odor encounters and a high likelihood of upwind turns ($p \sim 1$) when encounters are very frequent. We estimated the parameters from a maximum likelihood fit to the data, obtaining a distribution of parameters by performing the estimation on 500 distinct subsets of the measured data (Materials and methods). The distribution of estimated gains $\alpha$ clustered tightly around a mean of $\alpha = 0.242$ 1/Hz (*Figure 4F*), indicating that the parameter estimates were robust. Simulating this model with the mean of the estimated parameters closely reproduced the dependence of upwind orientation on encounter frequency (*Figure 4G*). Together, these findings indicated that in the spatiotemporally complex plume, odor encounters did not initiate reflexive upwind turning. Rather, odor encounters increased the likelihood that stochastically-occurring, saccadic left/right turns were directed upwind.

## Stopping and walking are stochastic events whose rates depend on odor encounter timing

Walking flies navigating spatiotemporally complex plumes stopped frequently (*Figure 1D*), and the rate of both stopping and starting depended strongly on the presence of odor (*Figure 2F*). To connect walk-stop transitions to individual encounters, we first calculated the likelihood to be walking or stopped during the 2 s after an encounter (*Figure 5—figure supplement 1A*). Walking flies were more likely to remain walking after an encounter (versus random times), while stopped flies were more likely to initiate a walk. Notably, even a single encounter was sufficient to initiate walks, and higher encounter frequencies biased this further (*Figure 5—figure supplement 1B*). This implicated both individual encounters and encounter history in decisions to walk or stop. In contrast, we found no change in walking speed following encounters (*Figure 5—figure supplement 1C*), even when encounter frequencies were appreciable (*Figure 5—figure supplement 1D*).

How does the sequence of encounters shape a fly's decision to walk or stop? After an odor encounter, flies walked for longer periods before stopping, compared to random (*Figure 5A*). Thus, encounters reduce stopping likelihood, and flies experiencing higher encounter frequencies walked for longer (*Figure 5B-C*). In addition, the time to stop following an encounter was the same, whether the encounter was isolated or part of a clump containing 3+ encounters in 1 s (*Figure 5A*). The times to stop were approximately exponentially distributed. We therefore modeled stop decisions as a Poisson process with a time-dependent stopping rate $\lambda_{w \to s}(t) = \lambda_{w \to s}(w(t))$, where $w(t)$ is the binary vector of encounter onset times. We considered various models for the dependency of the stopping rate on the encounter sequence $w(t)$. In the *last encounter* model, $\lambda_{w \to s}(t)$ drops to the same given value at each encounter, before decaying back to baseline with some characteristic time $\tau_s$ (*Figure 5D*). In the *accumulated evidence* model, $\lambda_{w \to s}(t)$ decreases further at every odor encounter, and therefore remains at a lower value when encounters are more closely spaced (*Figure 5H*). In the *encounter duration* model, $\lambda_{w \to s}(t)$ switches between a low value during encounters and a higher value during blanks (*Figure 5I*). These models contain various parameters dictating the baseline

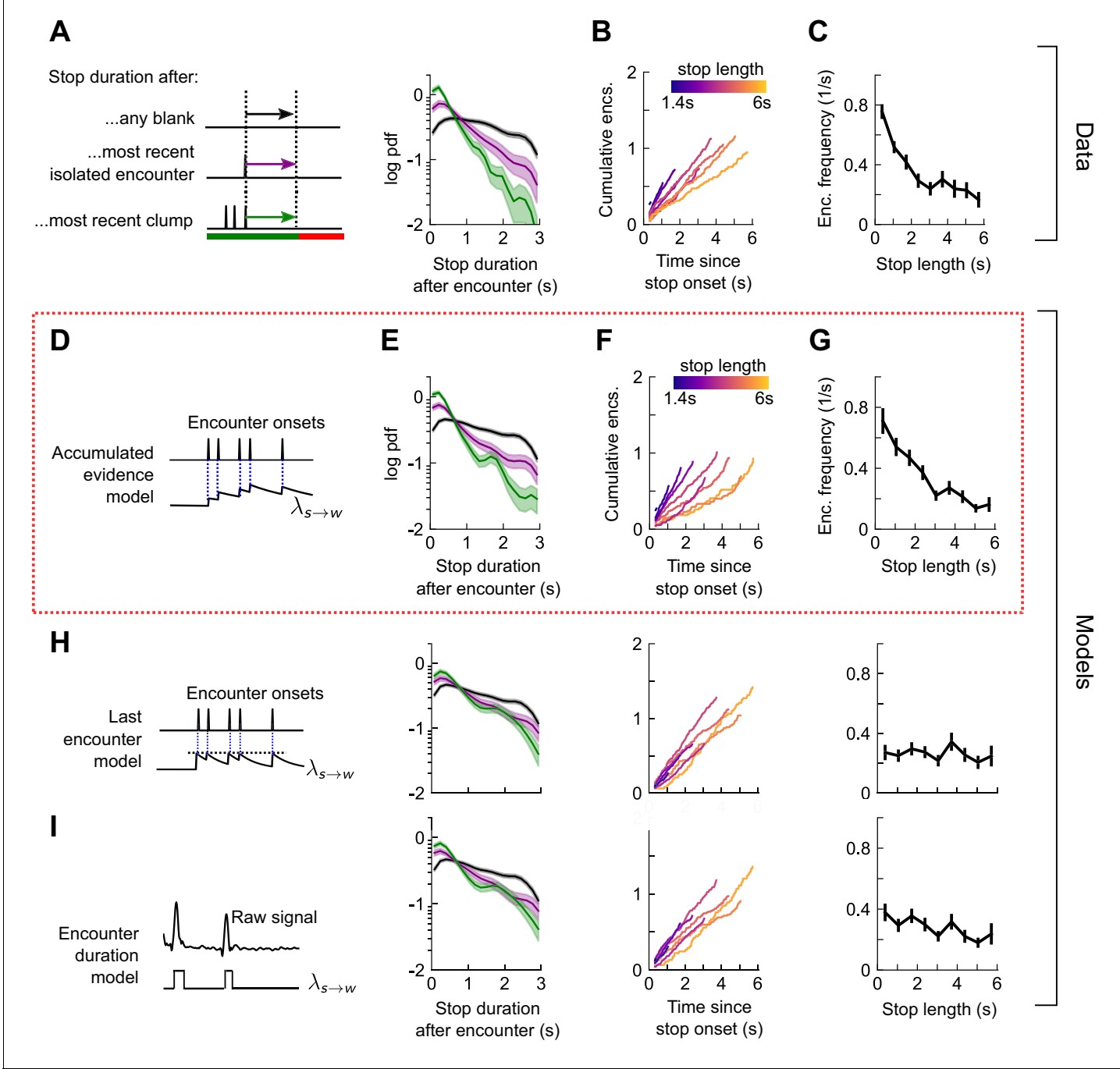

**Figure 6.** Walk decisions are stochastic events whose rate accumulates evidence from recent encounters. (A) Distribution of stop duration following any encounter, or the most recent encounter or clump before a walk. (B) Cumulative encounter counts since walk onset, for various stop durations. (C) Average encounter frequency versus duration of stop bout. (D) In the accumulated evidence model of walk decisions, $\lambda_{s\to w}(t)$ increases at every encounter onset, before decaying to baseline. This is modeled by $\lambda_{s\to w}(t) = \lambda_0 + \Delta\lambda \int e^{t-t'/\tau_w} w(t') dt'$. Median of estimated parameters are $\lambda_0$=0.29s$^{-1}$, $\Delta\lambda$=0.41s$^{-1}$, $\tau_w$=0.52s (*Figure 6—figure supplement 1*). (E-G) Analogs of A-C using data generated by the model. (H) Analogs of E-G for the last encounter model, in which the walk rate increases to a fixed value at each encounter before decaying to baseline. (I) Analogs of E-G for the encounter duration model, in which $\lambda_{s\to w}(t)$ switches between a high value during encounters and low value during blanks.

The online version of this article includes the following figure supplement(s) for figure 6:

**Figure supplement 1.** Distributions of estimated parameters for stop-to-walk models.

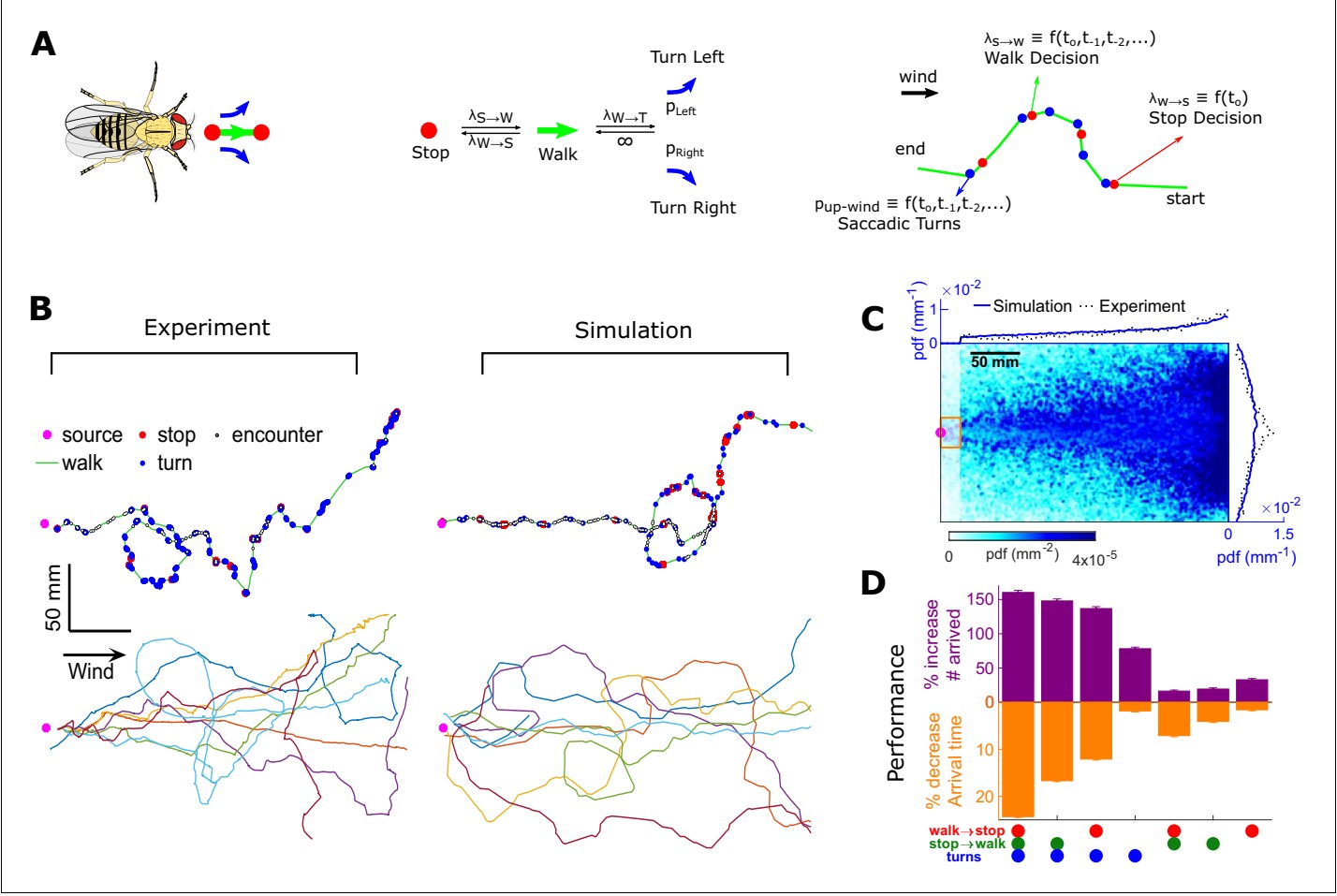

**Figure 7.** Agent-based simulation reveals navigational performance is significantly improved by encounter-modulated turn and walk decisions. (A) Agent-based simulation. Left: An agent has four behavioral states: stops, walks, and saccadic turns either left or right. Middle: Diagram of behavioral transitions. Right: Hypothetical trajectory of a virtual fly. Stop-to-walk rates and upwind turn probability depend on encounter history, whereas walk-to-stop rates depend only on the time of the last encounter. (B) Comparison of trajectories of real flies to those of virtual navigators in the complex plume. Top row: Representative trajectories. Bottom row: seven exemplary trajectories that reached within a 15 mm radius of the source. (C) Pdf of virtual flies (*n* = 10000 trajectories). Magenta: location of the source; blue curves: marginal pdfs over x- and y-direction for the simulation; dotted-black curves: marginal pdfs for the real flies (reproduced from *Figure 1K*). Very close to the source, the plume becomes ribbon-like, and real flies navigate this region by zigzagging around the slowly meandering ribbon as we have found in static ribbons (*Figure 1—figure supplement 1B*). In addition, flies tend to aggregate near the front of the arena near the odor inlet once reaching the source region. Since our models describe the navigation strategy in the interior of the plume, rather than this anterior near-source region, we have excluded the front 20 mm of the arena from the marginals. Red rectangle: borders used to determine whether a synthetic navigator reached to the source. (D) Purple: a fractional increase in the number of virtual flies that arrived at the source (borders of red box in C, same width in the y-direction as the one used in our static plumes [*Figure 1—figure supplement 1C*] and a length of 20 mm in the x-direction), $100(N_i - N_C)/N_C$, where $N_C$ is the number of control flies that arrived at the source, and $N_i$ is the number of flies arrived at the source for simulation condition $i$. Simulation conditions $i$ (x-axis) are distinguished by which behavioral components (turn, walk, and stop decisions) were removed. Stop and walk decisions were removed by setting the corresponding transition rate to their averages over all navigators in the full model (left-most bar). Turn decisions were removed by setting the upwind turn probability to its average. Orange: a fractional decrease in the arrival time, $100(T_c - T_i)/T_C$, where $T$ is the time to arrive to source. n = 762, 726, 694, 522, 341, 350, 389 trajectories. Error bars represent the SEM calculated by bootstrapping the data 30 times with replacement.

rates and timescales, which we fit to the data using maximum likelihood estimation. As in the turn model, we obtained a distribution of parameters by carrying out the estimation on 500 distinct subsets of the data – quantifying the robustness of each parameter (*Figure 5—figure supplement 2A*).

We found that the time-to-stop statistics were explained well by the last encounter model (*Figure 5E–G*) but not by the other models (*Figure 5H–I*). Our parameter fits indicate that at each encounter, the stopping rate drops to 0.17 s$^{-1}$, before rising with timescale 0.25 s to a background

rate of 0.78 s$^{-1}$. In the accumulated evidence model, the distribution of parameter estimates were broad, and often the parameters were estimated close to the imposed bounds (which ranged 2 orders of magnitude), suggesting that this model was not robust to the data (*Figure 5—figure supplement 2B*). The parameters of the encounter duration model were narrowly peaked, but the predictions were poor, so the model was incomplete (*Figure 5—figure supplement 2C*). In both these latter models, the distribution of walk durations following encounters was not higher than those following random times (compare black and dotted black lines in *Figure 5A E* to those in *Figure 5H–I*). Thus, our data indicate that flies continuously adjust their likelihood to stop while navigating and that the rate of stops decreases by a factor of nearly five at the onset of each encounter. This decrease in stop rate at each encounter is brief, less than 1 s, suggesting that the ongoing perception of frequent encounters retains the flies in an active, walking state, but when encounters are interrupted, stops are frequent.

Next, we quantified the rate of stop-to-walk transitions. In contrast to stops, the time to walk was significantly shorter following a clump of encounters than an isolated encounter (*Figure 6A*), implicating the history of encounters in walk initiation. In addition, the cumulative number of encounters received during a stop bout was rather independent of stop length, ~ 0.75-1.25 encounters for stops between 2 and 6 s long (*Figure 6B*). This observation would not rule out models in which the walking likelihood accumulated with every encounter, nor those in which the rate jumped to a large, fixed value at each encounter. Therefore, we modeled walk decisions with three models analogous to those used for the stop decisions. In the accumulated evidence model, the rate increases by the same amount at each encounter (*Figure 6D*), while in the last encounter model, the walk rate $\lambda_{s \to w}(t)$ increases to a set value at each encounter, before decaying to baseline (*Figure 6H*). In the accumulated evidence model, stopped flies receiving a clump would initiate walks sooner than those receiving a single encounter. In the encounter duration model, the rate switches between a higher value in encounters and a lower value in blanks (*Figure 6I*). The time-to-walk statistics were fit well by the accumulated evidence model (*Figure 6E-G*) but not the other two models (*Figure 6H-I*, *Figure 6—figure supplement 1*). The estimated baseline walking rate is $\lambda_0 = 0.29$ s$^{-1}$, so stopped flies will on average remain stopped for ~3 s if they receive no signal. This rate increases at each encounter by $\Delta\lambda = 0.41$s$^{-1}$, before decaying to baseline $\lambda_0$ with a timescale of 0.52s. Though our model predicts that a higher frequency of encounters will elicit an earlier walk, $\Delta\lambda$ is comparable to the base rate – more than doubling the transition rate – so even a single encounter is sufficient to elicit a walk, as observed in *Figure 6B* and *Figure 5—figure supplement 1B*. Together, this suggests that stopping forms a key component of the navigational strategy, and that stop and walk decisions are stochastic events whose rates of occurrence depend on the precise timing of recent encounters.

## Encounter-modulated decisions enhance navigational performance

To test how these behavioral algorithms individually affected navigational performance, we incorporated our findings into an agent-based simulation (*Figure 7A*). We simulated 10,000 individual virtual flies navigating using the turn, stop-to-walk, and walk-to-stop models that we found in our data. In these simulations, we calculated both the likelihood that agents reach the source as well as the time taken to do so. Virtual flies implementing all three encounter-modulated behaviors navigated largely in the plume cone (*Figure 7B*) and converged to the source (*Figure 7C*), similarly to real flies (10.5% of real flies and 7.8% of virtual agents reached within 15 mm of the source). Visually, the simulated tracks resembled the measured tracks, containing non-linear, circuitous routes toward the source, as well as wide loops (*Figure 7B*). To meaningfully test the contribution of the walk, stop, and turn decisions in effective navigation, we systematically replaced each time-dependent rate with its average, so that overall biases were retained but the dependency on encounters was not. Without encounter-modulated turning, adding stopping and walking decisions alone improved performance marginally (*Figure 7D*). With encounter-modulated turning present, however, the addition of either walk or stop decisions obeying our models both markedly increased the chance of finding the source and markedly reduced the search time. Together, this indicates a key benefit of encounter-driven stop-walk modulation when navigating spatiotemporally complex plumes.

## Discussion

Odor plumes can vary widely in spatiotemporal structure depending on the geometry of the surroundings and the nature of the airflow. In turbulent flows, the duration of odor encounters and blanks are power-law distributed, spanning a wide range of values from milliseconds to a few seconds long (*Celani et al., 2014*). While these flows become laminar near very smooth boundaries, the presence in natural terrains of obstacles, wind shifts, source motions, surface roughness, and boundary layer instabilities can cause smooth odor streams to break up into complex filaments (*Cardé and Willis, 2008*; *Murlis et al., 1992*; *Murlis et al., 2000*; *Riffell et al., 2008*). In our wind tunnel, we generate such perturbations by perturbing the laminar flow with stochastically alternating air jets near the upwind end. The key feature of this environment is that the statistics of the resulting odor patches are broadly distributed and approximate those in the atmospheric boundary layer (*Figure 1G–H*, *Figure 1—figure supplement 3*), while allowing us to image behavior and signal simultaneously.

The intermittent nature of turbulent odor plumes has inspired a number of theoretical navigational algorithms that treat odor signals as a train of event times (e.g. as in $w(t)$), ignoring encounter information about concentration and duration (*Balkovsky and Shraiman, 2002*; *Vergassola et al., 2007*). Indeed, information-theoretic analysis has indicated that precise measurements of odor concentration may confer less benefit than coarse measurements across space or time (*Boie et al., 2018*). In 'infotactic' searches (*Vergassola et al., 2007*), agents successfully navigate turbulent plumes by updating an internal spatial model of the plume structure, using only the arrival time of individual encounters. Analysis of insect flying trajectories (*Pang et al., 2018*) and *Caenorhabditis elegans* larvae crawling patterns (*Calhoun et al., 2014*) indicate that encounter-timing-driven infotaxis may form part of the navigation repertoire when concentration gradients are absent or difficult to measure.

Beyond theory, various experiments have shown that in intermittent plumes the frequency of encounters strongly shapes navigational behavior. The starkest indication of this in insect olfaction is the response of flying moths, *Cadra cautella* and *Heliothos Virescens* (*Mafra-Neto and Cardé, 1994*; *Vickers and Baker, 1994*; *Baker and Vickers, 1997*; *Carde and MafraNeto, 1997*), and walking moths, *Bombyx mori* (*Kanzaki and Sugi, 1992*), to pheromone plumes. In turbulent plumes and plumes pulsed at sufficient frequency, moths follow tight, narrow trajectories toward the source, whereas when the pulsing frequency is too low or the ribbon is static, they execute more zigzagging motion. To explain this, a model has been proposed in which an internal counterturning tendency is suppressed or reset by plume hits (*Kennedy and Marsh, 1974*; *Baker and Vickers, 1997*). A loose analogy could be made with our findings. Moths move crosswind and execute counterturns to find the plume, but once within the plume, high-frequency odor encounters cause them to suppress counterturns and surge upwind. Analogously, walking *Drosophila* move crosswind and execute a local search to get inside the plume but once inside a complex plume cone they execute random left/right saccades, with frequent odor encounters biasing these saccades upwind. In both cases, the timing and frequency of odor encounters suppress exploration and drive progress toward the source.

Like moths, flying flies navigating static odor ribbons counterturn back into them after passing through, effecting a similar upwind zigzag motion, though with smaller angles (*Budick and Dickinson, 2006*; *van Breugel and Dickinson, 2014*). An alternative explanation to the internal counterturning model in moth is that flies simply counterturn after losing the plume (*van Breugel and Dickinson, 2014*). The duration of encounters as flying flies pass through a static ribbon are brief – 10–250 ms – not unlike the encounters we measure in the plume used here. Further, due to the erratic zigzags of flies as they cross the ribbon, encounters are perceived somewhat randomly in time. Thus, it was suggested that since the statistics of perceived odor signals end up resembling those in turbulence, this plume loss-initiated counterturning might be a generic navigational strategy, occurring in spatiotemporally complex plumes as well (*van Breugel and Dickinson, 2014*).

At least for walking flies, we find that this is not the case. Turns occur stochastically, with rates independent of how long flies spend in the odor and the frequency of encounters (*Figure 4C*). But there is an important distinction between intermittency in flies crossing standing ribbons and those navigating dynamic plumes. In ribbons, intermittency is generated by animals' self-motion, creating a strong correlation between the likelihood of an odor encounter and spatial location. The location

of expected plume encounters is in this sense highly predictable, which makes counterturning an effective strategy. Within the cone subtended by our dynamic plumes, the frequency and duration of encounters are less correlated with location and direction, and can occur even when the fly is stopped. This makes the location of future hits less predictable (*Figure 1J*), so within the plume cone, reactive strategies such as counterturning might be ineffective.

An important finding here is that the duration of odor encounters plays no role in navigation (*Figure 3D–F*). This was unexpected, since a recent systematic quantification of navigation algorithms in walking *Drosophila* found that flies bias their orientation upwind by integrating odor concentration (*Álvarez-Salvado et al., 2018*). In that model, the concentration is normalized, so reorientations are accounted for primarily by the duration of the odor. However, the odor signals were relatively slow – pulsed from 0.1 to 1 Hz – giving encounter durations an order of magnitude larger than in the plume used here. This suggests that for rapid, intermittent signals, encounter frequency drives navigation, while for slower signals such as those expected in the boundary layer of a smooth surface, the duration of odor exposure matters. Effective navigation may therefore combine two important features of the temporal odor signal: its rectified derivative (giving encounter onset times) and it is integral (giving odor exposure time). Future studies interpolating between these extremes could elucidate if and how animals weight these two distinctly informative contributions.

A second important finding is that stopping forms a key component of the search strategy for walking flies (*Figures 5–7*). Stopping and waiting for encounters allows flies to receive odor encounters from dynamic plumes without wandering off-track or expending energy. We find that in deciding to walk, flies accumulate evidence from individual encounters, so walks are more likely following a clump of encounters than a single one. Theoretical work has shown that evidence accumulation from odor encounters can inform internal representations of plume structure to drive successful navigation in gradient-less plumes (*Vergassola et al., 2007*; *Calhoun et al., 2014*), an interesting possibility still to be examined. Filtering and integrating odor concentration drives navigation in odor plumes with longer encounters and less regularity (*Álvarez-Salvado et al., 2018*), suggesting that evidence accumulation – be it from odor duration or frequency – is a generic feature of olfactory navigation in a variety of environments. More work is required to understand the neural circuits and computations responsible for enacting stop and walk decisions. There is evidence that the transcription factor *FoxP* plays a role in value-based decision making, implicating these mutants as possible targets for future studies (*DasGupta et al., 2014*). Finally, encounter-elicited stopping might be unique to walking *Drosophila* and larvae (*Tastekin et al., 2018*), since remaining stationary is more difficult in flight. Still, the reflexive counterturns that flying *Drosophila* execute after losing a plume (*van Breugel and Dickinson, 2014*) do bear a loose resemblance to the increased stop rates following a drop in encounter frequency, so these decisions may have a common origin, but a different behavioral response.

Our visualizable signal is conventional smoke, a complex odor consisting of various aromatic compounds (*Figure 1—figure supplement 1A*). Gross fly behaviors in smoke are largely reminiscent of those in other known attractive odors, both in straight ribbons and complex intermittent plumes (*Figure 1—figure supplement 1*; *Figure 2—figure supplement 1*). Still, we expect differences in wind conditions, odor identity, and odor valence to modulate finer motor control in navigation (*Jung et al., 2015*). For example, the responses to the onset of the 15 s blocks of spatiotemporally complex signals were less pronounced in ACV than in smoke or ethyl acetate, with the latter being more similar to each other than to ACV (*Figure 2*, and *Figure 2—figure supplement 1*). Moreover, $Orco^{-/-}$ flies that lack major olfactory input, but are intact in $CO_2$ sensing, showed mild aversion around the center of the straight smoke plume, illustrating how different components contribute to the perception of the odor mixture.

It is surprising that despite the rich locomotive repertoire of walking *Drosophila*, a large part of their olfactory navigational strategy can be reduced to four actions – left turn, right turn, walk, and stop. A recent, systematic study of the locomotive structure of walking *Drosophila* in various windless odor environments has similarly found that behaviors fall into a limited number of states comprising a hierarchical hidden Markov model (*Tao et al., 2019*). While the identity of the odor and fly individuality affect the transition rates between these states, new states do not emerge in different conditions. These findings are consistent with ours. A natural extension would be to study how fly individuality and odor identity affect transition rates in our model, and which conditions would indeed require an extended behavioral space.

Finally, an important aspect not explored in our work is learning. The navigational algorithms we have found in the plume used here are shaped by odor information from the recent past, over time-scales no longer than a few seconds. Animals can learn odor landscapes over longer periods, by associating odor cues with spatial location. Desert ants, *Cataglyphis fortis*, have been shown to use learned olfactory scenes for homeward navigation in the absence of other directional cues (*Buehlmann et al., 2015*). Similarly, in mice, efficient foraging strategies can overtake an otherwise local gradient ascent strategy, if prior information about the odor scene is available (*Gire et al., 2016*). It is possible that the stochastic random walk strategies we observe here could be replaced with more stereotyped maneuvers if flies were sufficiently preconditioned to the environment. How the navigational strategies we have observed here are affected by conditioning, either with repeated trials or with reward feedback, provides a fruitful direction for future studies.

# Materials and methods

## Key resources table

| Reagent type (species) or resource | Designation | Source or reference | Identifiers | Additional information |
|---|---|---|---|---|
| Strain *Drosophila melanogaster* | Canton-S (CS) | John Carlson | NA | |
| Strain *Drosophila melanogaster* | Gr63A (w*;+;Gr63A$^{-/-}$) | Bloomington *Drosophila* Stock Center | Bloomington: RRID:BDSC_9941 | *Jones et al., 2007* |
| Strain *Drosophila melanogaster* | norpA$^7$ | Bloomington *Drosophila* Stock Center | Bloomington: RRID:BDSC_5685 | *Bloomquist et al., 1988* https://doi.org/10.1016/S0092-8674(88)80017-5(47) |
| Strain *Drosophila melanogaster* | Orco (w;+;orco[2]) | John Carlson | NA | https://doi.org/10.1038/nature11712 *Su et al., 2012* |
| Strain *Drosophila melanogaster* | Anosmic (Ir8a[1]; Ir25A[2]; Gr63A[1],Orco[1]) | Richard Benton | NA | *Ramdya et al., 2015* https://doi.org/10.1038/nature14024 |
| Chemical compound | Ethyl acetate | MilliporeSigma | CAT # 270989 | CAS # 141-78-6 |
| Chemical compound | Apple cider vinegar | Heinz | NA | |
| Software | Matlab, R2018a | MathWorks, Natick, MA | RRID:SCR_001622 | https://www.mathworks.com/products/new_products/release2018a.html |
| Software | Anaconda Distribution | Anaconda Inc, Austin, TX | NA | https://www.anaconda.com/ |
| Software | Python, 3.6.5 | Python Software Foundation | RRID:SCR_008394 | https://www.python.org/downloads/release/python-365/ |
| Other | Fly food | Archon Scientific, Durham, NC | NA | http://archonscientific.com/ |
| Other | Smoke wick | Regin Inc Oxford, CT, USA | S220 | Commercial wick to generate smoke |

## Experimental model and subject details
### Fly strains and handling
Flies were reared at 25°C and 60% humidity on a 12 hr/12 hr light-dark cycle in plastic vials containing 10 mL standard glucose-cornmeal medium (i.e. 81.8% water, 00.6% agar, 05.3% cornmeal, 03.8% yeast, 07.6% glucose, 00.5% propionic acid, 00.1% methylparaben, and 00.3% ethanol. Media was supplied by Archon Scientific, NC). All flies used in behavioral experiments were females. Newly eclosed flies were collected each day and placed in fresh vials. Females were then collected for starvation and placed in empty vials, 30–40 females in each vial, containing soaked cotton plugs at the

bottom and top. All flies were 5–9 days old and 3–4 days starved when experiments were performed. Experiments were carried out within 5 hr prior to the subjective sunset (i.e. 12 hr light turn off). All fly strains used in this paper are listed in the Key resources table.

Wild type CS and Orco$^{-/-}$ mutant flies were provided by John Carlson. Gr63a$^{-/-}$ (RRID:BDSC_9941) (*Jones et al., 2007*), and norpA$^7$ (RRID:BDSC_5685) (*Bloomquist et al., 1988*) mutant lines were purchased from Bloomington *Drosophila* Stock Center. Anosmic flies (Ir8a[1]; Ir25A[2]; Gr63A[1], Orco[1]) were provided by Richard Benton (*Ramdya et al., 2015*).

## Chemicals and reagents

Smoke wicks used to generate smoke were obtained from Regin Inc, Oxford, CT, USA (sku: S220). Ethyl acetate was purchased from Sigma-Aldrich (sku: S270989; CAS: 141-78-6), and apple cider vinegar, purchased from a grocery store, was made by Heinz.

## Method details

### Behavioral apparatus and stimulus delivery

Flies are introduced into an arena of size 300 mm (along wind) and 180 mm (across wind) with a depth of 10 mm (see schematic, *Figure 1A*). Flies walk unrestrained in this arena on glass surfaces, top and bottom, which were separated with acrylic walls and also bounded at the upstream and downstream end of the arena by straws and a plastic mesh, respectively. Experiments were recorded with an infrared (IR) sensitive camera (FLIR Grasshopper USB 3.0 NIR) in a dark room under IR illumination (850 nm). The recording rate was 30 Hz and 90 Hz for straight and intermittent plume experiments, respectively. The intermittent plume required a higher frame rate to track the dynamic smoke stimulus with sufficient resolution.

The behavioral apparatus operates as a wind tunnel. Active-charcoal filtered dry-air passes through the straws stacked at the upstream end of the arena and creates a laminar flow with a flow speed around 150 mm/s. Flow speed is measured by two methods: with an anemometer and by imaging and calculating the speed of a smoke plume in the laminar flow. The flow speed calculated with these two methods were similar (data not shown). The air and any odor carried with it were collected at the downstream end of the device with a vacuum hose loosely coupled to the device.

In order to deliver odorants into the behavioral chamber, clean air was passed over the headspace of pure odors placed in glass vials, and obtained odorized air was passed through a straw fixed at the center of the stack that creates the laminar flow. Variations in the odor dose was obtained by varying the ratio of odorized and clean air in the final flow delivered into the chamber. In the case of smoke, smoke generated by a burned wick (S220, Regin Inc) is accumulated in a 250 mL bottle for 20 s, and that bottle was used as the smoke-odor source.

Straight plumes were obtained by simply matching the odorized air flow speed to laminar flow speed. In order to generate intermittent plumes, air flows (flow speed:~1500 mm/s) perpendicular to the laminar flow were injected into the arena near the upstream end of the device. The injected air flows were randomly alternated between the left and right side of the arena with 100 ms correlation time.

### Experimental protocol

Starved flies, between 30 and 40 in number, were aspirated into the arena all in once while the wind was on and allowed 1 min to acclimate to their environment before the experiment. This odor was on during the whole experiment, which lasted 90 s (unless otherwise noted), however, it took several seconds,~5 s, to pass through the tubing and enter into the arena. In pulsed intermittent plume experiments, while the random lateral air injections persisted, the odor was alternatively turned on and off in 15 s blocks. Experiments were repeated three times on the same flies, leaving 3 min intervals in between. The humidity and temperature of the room is logged for each experiment. Room temperature was stable at 24.2 ± 0.3 °C. Although the room humidity varied between 25% and 43% (average: 32.5 ± 2.5%) depending on the season of the year, flies were tested in dry air flow which had humidity close to zero.

To minimize contamination caused by odor molecules sticking to inner surfaces of the experimental apparatus we took the following precautions: First, we used only PTFE Teflon (a hydrophobic,

chemically inert, and low friction material that minimizes odor sticking) tubes to deliver odors. Second, we kept experiment durations relatively short (less than 2 min) which eliminated continuous accumulation of residual odorants on the glass surfaces. Third, the main laminar flow was kept flowing for 3–5 min in between trials to wash away residual odors on the glass surfaces and acrylic side walls. Fourth, flies in close vicinity (~3 mm) to the plastic mesh at the downwind end of the apparatus and acrylic side walls were not tracked and therefore not included in the analysis. Fifth, glass surfaces and acrylic side walls were wiped clean with Windex and the downwind plastic mesh was replaced with a clean one before each experiment. Finally, the whole stack of straws and tubing were replaced with clean ones before a different odor was tested.

Our experimental protocol starts with recording flies in clean air for 90 s before the odor is presented for the first time. This control recording is followed by the experiment that involves odor release and lasts for 90 s. This experiment is repeated three times with the same flies allowing 3 min clean air periods in between trials, before the protocol is repeated on different flies. To assess whether possible residual odors in the arena affected navigation, we compared the behavior of flies in clean air to that of in odorized air. *Figure 1L* shows that flies in clean air do not accumulate near the upstream straws or the side walls, even though this is where odors are likely to stick more.

## Quantification and statistical analysis

All analysis was performed using custom-written Matlab and Python scripts.

## Fly tracking and signal estimation

### Fly tracking

Fly tracks were extracted by analyzing recorded videos. Non-uniform illumination in the arena was corrected by dividing each frame pixel-by-pixel with a flat field image. The flat field image was obtained as follows: (a) median smoothing (filter size ~ 8 mm) the image of the arena free of flies, (b) fitting a 5th-order polynomial to the smoothed surface, (c) normalizing the fitted surface with the mean value. Following the flat-field correction, each frame was thresholded and binarized. In the binarized images, objects with an area larger than one square millimeter were registered as flies. The positions ($x$ and $y$ coordinates) and orientations ($\theta$) of the flies were obtained by finding the centroids and major axis of these regions (Matlab regionprops). Tracks were assembled by linking each centroid in a frame to the closest one in the consecutive frame. Whenever two or more flies interacted (i.e. passing over each other on opposite surfaces or collision), water-shedding was used to resolve the identity of the flies. If water-shedding failed, fly positions were predicted based on their recent average velocities until they separate, and identities were then assigned by comparing the predicted positions with the positions right after separation. If this failed, the track was flagged and eliminated from subsequent analysis.

### Signal estimation

The signal $s(t)$ that flies experience along their path was estimated by calculating the mean smoke intensity in a virtual antenna fixed in front of the flies' head (*Figure 1B* inset). The virtual antenna was as wide as the fly (i.e. 1.72±.24 mm), and its length was set to one-fifth of the fly minor axis (average: 0.46±.08 mm). The distance between the virtual antenna and fly head (i.e. 1.24±.22 mm) was optimized by minimizing the overlap between the virtual antenna and dilated fly body (dilation number: 7). Instances in which the virtual antenna overlapped with another fly or its reflection were flagged as unreliable and eliminated from all analyses that required odor information. The accuracy of our automated detection system was validated by comparing its output to the output of manual annotation. Four videos were manually annotated, frame by frame, by two researchers for the validity of the signal values in the virtual antenna of all flies. The total number of annotated time points was 166,411. This comparison revealed that only 0.97% of all data points were assigned as false positives by our automated software, whereas the fraction of trajectories that were assigned as false negatives was 6.05%.

## Encounter detection

Due to shot noise, signal values were above zero. In order to calculate the mean background signal value, for each recorded video, we fitted a Gaussian to the distribution of the signal values of all flies and set the mean signal background to the mean of the Gaussian. The signal was then thresholded with a threshold value, equal to the sum of the mean and 2.5 SD of the background signal, and binarized for encounter detections. Instances with signal values higher than the threshold are identified as encounters, and blanks are defined as the intervals between encounters.

## Error estimation in the timing of encounter detection

The difference in the location of the virtual antenna from that of the actual fly antenna introduces some uncertainty in signal timing. This uncertainty depends on wind speed and direction, and we estimated it as 8.3±1.5 ms and -8.3±1.5 ms for downwind and upwind oriented flies, respectively. Further uncertainty in timing is introduced by our sensory threshold, which we chose as 2.5 SD above the mean camera shot noise. To estimate this uncertainty, we considered flies navigating a static smoke ribbon. We calculated the lateral distance from the plume at which flies counterturn back into the ribbon after passing through it (counterturn positions were found using Matlab's 'findpeaks' function), and compared it to the iso-line of the smoke intensity used for the sensory threshold ($2.5\sigma$; *Figure 1—figure supplement 2A*). The average counterturn positions along the ribbon (calculated using Matlab's LOWESS smoothing function) aligned with the iso-line of the minimum smoke intensity captured by the camera (*Figure 1—figure supplement 2B*). However, they differed from the iso-line of our $2.5\sigma$ sensory threshold by 1.94 mm at the downstream end of the device, giving an upper bound of 13 ms uncertainty in encounter timing. Combined with uncertainty from the virtual antenna locations, we estimate the overall error in encounter timing to be less than 25 ms.

## Analysis of behavioral data

### Smoothing of measured behavioral time traces

From the procedures described above, we obtained for each fly trajectory the position (x and y coordinates) and the orientation ($\theta$) of the fly, together with the signal s in the virtual antenna. To remove measurement shot noise from the fly tracking and signal estimation, we filtered each of these quantities with a Savitsky-Golay filter with k-order polynomial and m-length windows, where k = 4 and m = 9. Taking the derivative of the fitted piecewise Savitsky-Golay polynomials for x and y gives us smoothed velocity components $v_x$ and $v_y$, respectively, from which we obtain the speed $v = \sqrt{(v_x^2 + v_y^2)}$. Similarly, the time rate of change of orientation, $\omega = d\theta/dt$, was found by converting $\theta$ to x-y components on the unit circle, smoothing each of these with the same Savitsky-Golay filter, taking their first derivative (which is the analytical derivative of the smoothed polynomials), and then converting back to polar coordinates. This last conversion was done using $\omega = d\theta/dt = -\dot{\theta}_x/\theta_y = \dot{\theta}_y/\theta_x$, the appropriate equation being chosen if either $\theta_x$ or $\theta_y$ were too small. This procedure removes issues with the branch cut at $\theta = 0$, which would arise if $\theta$ were differentiated directly. Finally, upwind orientation $\theta_+$ was determined by reflecting all angles over the polar x-axis to the range 0-180 degrees. All subsequent analyses used these smoothed time traces.

### Determination of stop, turn, and encounter events

From these smoothed quantities, all binary events – stops, turns, and encounters – were determined in the same way, using a thresholding technique that minimizes false detections. Specifically, the onset of an event is said to occur when the quantity is above threshold, but only if the time above the threshold is longer than some set duration. This prevents false detections of artificially short events that may arise from measurement fluctuations. The same requirement is enforced for the event offset: the quantity must drop below threshold for a sufficient time. For encounter instances, the threshold was set to $2.5\times$ the standard deviation of the background signal, when fit to a Gaussian. For stops, the threshold was set at $v = 2$ mm/s, and for turns the threshold was set to $|\omega| = 200$ deg/s. The minimum duration for stops was set to 300 ms, for turns was set to 20 ms, and for encounters was set to 50 ms.

To verify that the discrete nature of turns was not an artifact of turn detection thresholds, we used the following analysis, whose results are presented in *Figure 4—figure supplement 1*. Turn events were first detected for a given threshold on absolute angular velocity, $|\omega|$. A window of length $\Delta T$ is defined around the midpoint of each turn, over which the total orientation change $\Delta\theta$ of the fly can be defined (*Figure 4—figure supplement 1A*, top plot). If angle changes are indeed discrete, then the distribution of $\Delta\theta$ would increase with $\Delta T$ for small $\Delta T$, but would level out asymptotically to a bimodal distribution for sufficient $\Delta T$, corresponding to the maximum length of a stereotypical turn. For different choices of $|\omega|$ (different colored plots in *Figure 4—figure supplement 1B*), darker shades correspond to $\Delta\theta$ distributions for increasing window lengths $\Delta T$. The distributions are invariant once the window is long enough, showing that for sufficiently long time window, the turns are discrete. The positive peak of the distributions is plotted as a function of $\Delta T$ (*Figure 4—figure supplement 1C*). This peak levels off above $\Delta T$ = 150 ms, for all turn thresholds larger than 150 deg/s. As such, we chose a threshold of 200 deg/s for turns, verifying visually that false positives and false negatives were minimized.

## Fly distributions in the arena

The probability density function (pdf) of fly distribution in the arena (*Figure 1K–L*, *Figure 7C*, *Figure 1—figure supplement 1C*, *Figure 1—figure supplement 1E*, *Figure 1—figure supplement 1G*, and *Figure 1—figure supplement 1I*) is estimated by calculating the histogram (count) of fly positions for each unit area ($\Delta X = \Delta Y$ = 1 mm), and normalizing that histogram with the total count of fly positions. Areas with zero fly visits are indicated with white color in the color scale. All positions for all time points throughout the experiment are used to calculate the histogram, and therefore the estimated pdf represents the cumulative fly walking behavior. Trajectories shorter than 5 s, with a mean speed less than 2 mm/s, and with a total displacement less than 50 mm are eliminated.

## Calculation of stop-to-walk and walk-to-stop transition rates

To calculate the transition rate from a stop to a walk $\lambda_{s \to w}$ (*Figure 2F*), we considered each part of the trajectory containing one stop followed by one walk. In this range, we set $\lambda_{s \to w} = T_{stop}^{-1}$, where $T_{stop}$ is the duration of the stop. Likewise, we get the transition rate from walk to stop, $\lambda_{w \to s}$, by considering every trajectory snippet containing one walk followed by one stop. In this range, $\lambda_{w \to s} = T_{walk}^{-1}$. Thus, the walk-to-stop rates update their values at the onset of every stop, and vice versa for walks.

## Numerical methods

### Statistical tests

Unless noted otherwise, all error bars represent standard error of the mean. Stars in the manuscript indicating significance are *$p<0.05$, **$p<0.01$, ***$p<0.001$. Unless otherwise noted, for estimating SEM through bootstrapping, 5000 resamples were chosen, with resample size half of the original data size.

The trivariate linear regression we use to determine the dependence of upwind orientation $\theta_+$ on $W_{dur}$, $W_{freq}$, and $W_{conc}$ (*Figure 3F*) can exhibit issues of multicollinearity when the two regressors are correlated. This can produce erroneous and non-robust parameter estimates. To check for this, we ensured that the moment matrix $X^T X$, where $X$ is the matrix of observations, was not ill-conditioned. We found conditions numbers less than 2, indicating no ill-conditioning. We also ensured that the estimated parameters were robust to different subsets of the data. We found that the parameter ranges for these subsets are statistically indistinct from those of the full dataset.

### Calculation of encounter frequency and encounter duration

Calculation of the odor encounter frequency $W_{freq}$ was done by convolving the binary vector of encounter times $w(t)$, which contains a '1' at the onset of every encounter and 0s elsewhere, with an exponential filter:

$$W_{freq}(t) = \int_{-\infty}^{t} e^{\frac{t-t'}{\tau}} w(t') dt'$$

where $\tau$ is set to 2 s. This timescale was chosen to optimize the trends but our findings were robust to this value. Similarly, the encounter duration $W_{dur}$ was calculated by convolving the binary vector of encounter exposure, $d(t)$ which contains a '1' whenever the fly is within a encounter and 0 s elsewhere, with the same exponential filter:

$$W_{dur}(t) = \int_{-\infty}^{t} e^{\frac{t-t'}{\tau}} d(t')dt'$$

Finally, odor concentration was calculated by convolving the raw signal $s(t)$ (an 8-bit integer representing the intensity of the imaged smoke signal) with the same exponential filter:

$$W_{conc}(t) = \int_{-\infty}^{t} e^{\frac{t-t'}{\tau}} s(t')dt'$$

## Modeling
### Stochastic models of stereotyped turning
Turns are modeled as a homogeneous Poisson processes with rate $\tau_T^{-1} = \lambda_T = 1.3$ turns/s, determined by fitting the data to an exponential distribution (*Figure 4C*). While we did not find a statistically significant dependence of the turning rate on encounter frequency or duration, we allowed for this dependence in fitting the model to the data (below). When a turn occurs, its magnitude and direction are also sampled from probability distributions. The absolute value of the turn angle is chosen from a normal distribution with mean $30^o$ and standard deviation $10^o$, representing the two peaks in the bivariate distribution in *Figure 4A*. The turn direction is a binomial variable with probability $p_T(w(t))$ that the direction is upwind, where the encounter onset times $w(t)$ are defined above. Guided by the observation that the upwind turning probability increases with encounter frequency, we set $p_T(w(t))$ as:

$$p_T(w(t)) = (1 + e^{-\alpha x})^{-1}, \text{where} x = \int_{-\infty}^{t} e^{\frac{t-t'}{\tau}} w(t')dt' = W_{freq}$$

where the filter timescale $\tau$ is set to 2 s as above.

### Stochastic 'accumulated evidence,' 'last encounter,' and 'odor duration' models of walk-stop transitions
Walk-to-stop transitions are modeled as an inhomogeneous Poisson process with rate $\lambda_{w \to s}(t) = \lambda_{w \to s}(w(t), d(t))$, where $w(t)$ and $d(t)$ are defined above. The distinguishing feature between the three models we test, the last encounter model, the accumulated evidence model, and the encounter duration model, is their functional dependence on $w(t)$ and $d(t)$.

In the last encounter stopping model, we have:

$$\lambda_{w \to s}(t) = \lambda_0 + \Delta\lambda e^{-\Delta T(w(t))/\tau_s}$$

where $\Delta T(w(t))$ indicates the time since the most recent encounter. In the accumulated evidence stopping model, we have:

$$\lambda_{w \to s}(t) = \lambda_0 + \lambda_1 / \left(1 + \lambda_2 \int e^{\frac{t-t'}{\tau_s}} w(t')dt'\right)$$

Finally, in the encounter duration model, we have:

$$\lambda_{w \to s}(t) = \lambda_w d(t) + \lambda_b(1 - d(t)).$$

In the encounter duration model, therefore, the rate is $\lambda_w$ during encounters and $\lambda_b$ during blanks. The rates for the stop-to-walk transitions are analogous. For the last encounter model, we have:

$$\lambda_{s \to w}(t) = \lambda_0 + \Delta\lambda e^{-\Delta T(w(t))/\tau_w},$$

for the accumulated evidence walking model, we have:

$$\lambda_{s \to w}(t) = \lambda_0 + \Delta\lambda \int e^{\frac{t-t'}{\tau_w}} w(t') dt',$$

and for the encounter duration model, we have:

$$\lambda_{s \to w}(t) = \lambda_w d(t) + \lambda_b (1 - d(t))$$

## Turn model parameter estimation

In our turning model, the gain parameter $\alpha$ must be estimated from the data. While we measured the turn rate $\tau_T$ from experiment and found no significant dependence on encounter duration or frequency (**Figure 4C**), we nevertheless allow it to be a free parameter with a linear dependence on both. The full turn rate $\lambda_F = \tau_F^{-1}$ is therefore:

$$\lambda_F = \tau_T^{-1} + \tau_{freq}^{-1} W_{freq}(t) + \tau_{dur}^{-1} W_{dur}(t).$$

The turning model thus contains four unknown parameters: $\Theta = \alpha, \tau_T, \tau_{freq}, \tau_{dur}$, which we obtain using maximum likelihood estimation. The expression for the maximum likelihood depends on various conditional probability distributions, which we now discuss. The measured data are orientation changes $d\theta_i$ at each time step during which the fly is walking. These angle changes arise either during straight bouts, where the angle changes consist of small zero-mean jitter, or during turns. At any given time $i$, the probability of angle change $d\theta_i$ is:

$$p(d\theta_i; \Theta) = \sum_X p(d\theta_i | X; \Theta) P(X; \Theta),$$

where $X$ denotes the three possible behaviors: upwind turns, downwind turns, and straight bouts. The probability that a turn is upwind, $p(up; \Theta)$, is the probability of a turn times the probability that the turn is upwind, given by our turn model above. This gives:

$$p(up; \Theta) = \Delta t \, \lambda_F e^{-\Delta t \lambda_F} p_T(w(t)).$$

$$p(down; \Theta) = \Delta t \, \lambda_F e^{-\Delta t \lambda_F} (1 - p_T(w(t))),$$

where $\Delta t$ is time step of the measured data. The distribution of upwind angle changes, $p(d\theta_i | up; \Theta)$, is assumed Gaussian with $\sigma = 10^o$ and mean $+30^o$ (for $\theta_i$ between 0 and 180°) and $-30^o$ (for $\theta_i$ between 180° and 360°). These values represent the mean and spread of the two peaks in the measured $d\theta_i$ during turns (**Figure 4A**). The distribution of downwind angle changes $p(d\theta_i | down; \Theta)$ is the same, but with opposite means. The likelihood of a straight walk is:

$$p(straight; \Theta) = e^{-\Delta t \lambda_F}.$$

The jitter for straight bouts was measured to be 0.22° per time step, whereby $p(d\theta_i | straight; \Theta)$. is assumed normal with standard deviation 0.22° and mean 0. With these distributions, the likelihood function of parameters $\Theta$ given the measured data $d\theta_i$ reads:

$$L(\Theta) = \prod_{i = \text{upwind turns}} p(d\theta_i | up; \Theta) p(up; \Theta) \times$$

$$\prod_{i = \text{downwind turns}} p(d\theta_i | down; \Theta) p(down; \Theta) \times$$

$$\prod_{i = \text{straight walks}} p(d\theta_i | straight; \Theta) p(straight; \Theta)$$

The first product contains only times at which the flies turn upwind, etc. The parameters were estimated by maximizing the logarithm of $L(\Theta)$. The optimization was performed using the limited-memory Broyden–Fletcher–Goldfarb–Shanno optimization algorithm (L-BFGS). In our implementation of this method, we only provide the cost function (the log-likelihood), and the gradient is computed numerically with finite differences. L-BGFS approximates Hessians from function and gradient

evaluations in previous iterations, using this to steer the estimate toward the function minimum. To get an idea of the spread and robustness of parameter values, we perform this optimization for 500 distinct subsets of the data, where each subset contains 20% of all measured trajectories, randomly chosen. We find that the distribution of the 500 estimated $\alpha$ and $\tau_T$ are highly peaked around a mean of 0.046 Hz$^{-1}$ and 0.75 s, respectively, suggesting that these values are robust to various subsets of the experimental data. The estimated coefficients $\tau_{freq}$ and $\tau_{dur}$ are small (medians 0.008 and 0.07, respectively, which would each contribute <3% to the base timescale $\tau_T$, given the encounter statistics in our intermittent plume), and span both negative and positive values, suggesting that they are not robust. These are in accordance with the lack of dependence in turn frequency on encounter duration or frequency (*Figure 4C*).

## Stop and walk models parameter estimation

The likelihood function for stop and walk models is similar to the turn model. Since the walk-to-stop and stop-to-walk rates are distinct inhomogeneous Poisson processes, with distinct functional forms $\lambda_{s \to w}(t)$ and $\lambda_{w \to s}(t)$ for the rates, respectively, we write the likelihood function for walk-to-stop transitions $L_{w \to s}(\Theta)$ only here. The other, $L_{s \to w}(\Theta)$, is analogous.

The likelihood function for walk-to-stop transitions is:

$$L_{w \to s}(\Theta) = \prod_{walk\ times} p(walk; \Theta) \prod_{stop\ onsets} p(stop; \Theta)$$

The first product runs over all times that the fly is walking, while the second product contains only the initial point of each stop bout, which represents the decision to stop, given that the fly is walking. Like the turning model, the rates follow a Poisson process, giving:

$$p(walk; \Theta) = e^{-\Delta t \lambda_{w \to s}}$$

$$p(stop; \Theta) = \Delta t\, \lambda_{w \to s} e^{-\Delta t \lambda_{w \to s}},$$

where the dependence on the parameters $\Theta$ enters through $\lambda_{w \to s}$. Parameters are estimated as described above by minimizing the logarithm of the likelihood. Rather than optimizing for $\tau_w$ or $\tau_s$, both of which enter the cost function in the denominators (introducing artificial singularities in the derivatives), we optimize for their inverses, which enter linearly in the exponents.

Since we are fitting stochastic dynamical models (Poisson processes generated by a nonlinear rate function with ODE dynamics), our models naturally allow us to cross-validate by generating predictions to novel stimuli. Cross-validation is a natural check for overfitting and is particularly suited to dynamical models (*Gábor and Banga, 2015*; see *Generation of stop and walk model statistics* below). In nonlinear models with hidden states, near-perfect fits to the data can often produce highly inaccurate predictions when applied to novel data. These inaccuracies are masked by the unobservability of the hidden states but can be revealed by cross-validation (*Kadakia et al., 2016*; *Ye et al., 2015*). In our case, the models were shown to give quite distinct predictions from the estimated parameters. Cross-validation, therefore, provides a strong test for model comparison in our case.

Alternatively, Akaike Information Criterion (AIC) and Bayesian Information Criterion (BIC) are both statistical tests for model comparison that combine the log likelihood with a contribution from the number of model parameters. If we calculate the AIC and BIC for our models, for all 500 subsets of the data for which we carried out parameter estimates, we do not find a statistical distinction between the AIC or BIC distributions (among these 500 subsets) for either the set of walk-to-stop models or the set of stop-to-walk models. This can be explained by the following reasoning. The number $k$ of model parameters is largely similar between our models, and both $k$ (in the case of AIC) or $k\ln(n)$ (where $n$ is the number of data points, in the case of BIC) is significantly smaller than the log likelihood $L$, which therefore dominates the AIC/BIC values. $L$ is a sum over all timepoints but odor encounter frequency is not evenly distributed among these timepoints: low encounter timepoints are far more common than high-frequency timepoints. Thus, $L$ is strongly biased by lower encounter frequency data. In principle, we would like the model predictions to match equally well against the data for all encounter frequencies (e.g. as in *Figure 5D–I* and *Figure 6D–I*, which span low and high encounter frequencies). This would not manifest in $L$, and thus would not manifest in the AIC or BIC criteria.

On the other hand, using cross-validation, we find that the predictions are quite distinct for the models (*Figure 5D–I* and *Figure 6D–I*). In addition to the reasoning given above, cross-validation requires fewer assumptions than AIC/BIC – there are no Laplace approximations (Gaussian approximation near the optimal parameter) being performed, nor is it derived in the limit of infinite data. Finally, for all our models, the log likelihoods are highly nonlinear, so the Gaussian assumption needed for the Laplace approximation in the AIC and BIC is not well-satisfied.

## Generation of turn model predictions

To generate the upwind orientation versus encounter frequency plot predicted by the model estimates (red line in the second plot of *Figure 4G*), we chose from the 500 sets of estimated parameters Θ the median of each parameter. Synthetic $\theta(t)$ traces were generated by applying the model to a synthetic random encounter trace which contained encounter frequencies spanning a range from very low (< 0.1 Hz) to high (10 Hz), mimicking the range of frequencies encountered in the data.

## Generation of stop and walk model statistics

To generate the statistics in Figures 5-6, we chose from the 500 sets of Θ the median of each parameter. Synthetic stops and walks were then generated from the aggregated measured $w(t)$ of all trajectories by simulating a Poisson process with the corresponding rate functions. This procedure generated a synthetic time series of stop events, given that the fly is walking. From this vector, we calculated time-to-stop for all cases as in the measured data: (i) all encounters, (ii) all isolated encounters, and (iii) all isolated clumps. A similar procedure was used to generate a vector of walk events, given that the fly is stopped. From this vector, we calculated time-to-walk for analogous cases: (i) all encounters, (ii) all isolated encounters, and (iii) all isolated clumps.

## Agent-based simulation

Virtual agents navigated the same arena and the same intermittent smoke environment as experimental flies. Each simulation batch was run using 10,000 agents, initialized randomly along the back, downwind wall of the arena. The simulation ran for 11,690 time steps at a timestep corresponding to the same sampling rate as the recorded videos, 90 frames per second. The agents were assigned a walking speed equal to the mean walking speed of experimental flies (10.1 mm/s), and their turns were assumed to be instantaneous. Turn angles were drawn from a normal distribution with mean 30° and standard deviation 10°. The navigators were given elliptical virtual antennas, the centers of which were located approximately 14 pixels (2.16 mm) in front of the fly centers. The elliptical antennas had a semi-major axis of five pixels (0.77 mm) and semi-minor axis of 1.5 pixels (0.23 mm); the semi-major axis was oriented perpendicular to the agent's orientation vector. To avoid spurious detections arising from jitter in the signal, the virtual navigators registered an encounter only if they had not encountered one in the past 100 ms.

To remove individual navigational components, the corresponding rate for that component was set to an average rate from the full model. For example, to remove stop decisions, the walk-to-stop rate was set to the average. The average stop-to-walk rate was determined by fitting the distribution of stop durations from all 10,000 trajectories in the full model to an exponential distribution; the fitted timescale was used as the average rate. The average walk-to-stop rate was determined similarly from the distribution of walk durations. The average probability of turning upwind was defined as the fraction of turns directed upwind by all 10,000 trajectories in the full model.

## Data and code availability

The fly lines used in this study are available upon request. The data are available at https://doi.org/10.5061/dryad.4j0zpc87z and the scripts used to perform the experiments, track flies, and extract relevant behavioral data are available on https://github.com/emonetlab/fly-walk (copy archived at swh:1:rev:6a9266effbdc305c2e6177a7b6786e295cb48a2c) and scripts used to run simulations are available on https://github.com/emonetlab/fly-walk-sims (copy archived at swh:1:rev:be9b-b7a93eb4963ca0515144940694412304f633). We thank the following people for making their Matlab scripts, utilized for generating plots in this work, freely available: Ben Mitch, Panel; Holger Hoffman, Violin; Kelly Kearney, legendflex; Yair Altman, export_fig; David Legland, geom2D; Rob Campbell, shadedErrorBar.

## Acknowledgements

We thank Viraaj Jayaram for his help in running agent-based simulations, John Carlson for providing several fly lines used in this study and for giving us access to instruments in his lab, Richard Benton for providing anosmic flies, Francesco Carbone and Kevin Gleason for performing GCMS analysis on the smoke. Srinivas Gorur-Shandilya helped write the fly tracking code. We thank Pedro Cisneros and Abhishek Sethi for their help writing scripts and annotating experimental videos. MD was supported by the Program in Physics, Engineering and Biology at Yale University. MD and TE were partially supported by the Allen Distinguished Investigator Program (grant 11562) through The Paul G Allen Frontiers Group. TE was partially supported by NIH R01GM106189. NK was supported by a postdoctoral fellowship through the Swartz Foundation and by postdoctoral fellowship NIH F32MH118700. HDA was supported by NSF DBI-1755494. DAC was supported by NIH R01EY026555, a Searle Scholar Award, a Sloan Fellowship in Neuroscience, and the Smith Family Foundation. This research was supported in part during a visit to KITP by NSF Grant No. PHY-1748958, NIH Grant No. R25GM067110, and the Gordon and Betty Moore Foundation Grant No. 2919.02

## Additional information

### Funding

| Funder | Grant reference number | Author |
| --- | --- | --- |
| Allen Foundation | 11562 | Mahmut Demir<br>Thierry Emonet |
| National Institutes of Health | R01GM106189 | Thierry Emonet |
| National Institutes of Health | F32MH118700 | Nirag Kadakia |
| Swartz Foundation | | Nirag Kadakia |
| National Science Foundation | DBI-1755494 | Hope D Anderson |
| National Institutes of Health | R01EY026555 | Damon A Clark |
| Yale University | Program in Physics, Engineering and Biology | Mahmut Demir |
| National Science Foundation | PHY-1748958 | Nirag Kadakia<br>Thierry Emonet |
| National Institute of General Medical Sciences | R25GM067110 | Nirag Kadakia<br>Thierry Emonet |
| Gordon and Betty Moore Foundation | 2919.02 | Nirag Kadakia<br>Thierry Emonet |

The funders had no role in study design, data collection and interpretation, or the decision to submit the work for publication.

### Author contributions

Mahmut Demir, Conceptualization, Resources, Data curation, Software, Formal analysis, Validation, Investigation, Visualization, Methodology, Writing - original draft, Writing - review and editing; Nirag Kadakia, Resources, Data curation, Software, Formal analysis, Validation, Investigation, Visualization, Methodology, Writing - original draft, Writing - review and editing, Statistical analysis and modeling; Hope D Anderson, Software, Formal analysis, Visualization, H.A. performed the agent-based simulations and their analysis; Damon A Clark, Formal analysis, Methodology, Writing - review and editing; Thierry Emonet, Conceptualization, Formal analysis, Supervision, Funding acquisition, Validation, Methodology, Writing - original draft, Project administration, Writing - review and editing

### Author ORCIDs

Mahmut Demir ⓘD https://orcid.org/0000-0002-3278-7843
Nirag Kadakia ⓘD https://orcid.org/0000-0001-9978-6450

Damon A Clark (iD) http://orcid.org/0000-0001-8487-700X
Thierry Emonet (iD) https://orcid.org/0000-0002-6746-6564

## Decision letter and Author response

Decision letter https://doi.org/10.7554/eLife.57524.sa1
Author response https://doi.org/10.7554/eLife.57524.sa2

## Additional files

### Supplementary files
• Transparent reporting form

### Data availability

The data is be available at https://doi.org/10.5061/dryad.4j0zpc87z and the scripts used to perform the experiments, track flies, and extract relevant behavioral data and run simulations are available on https://github.com/emonetlab/fly-walk (copy archived at https://archive.softwareheritage.org/browse/revision/6a9266effbdc305c2e6177a7b6786e295cb48a2c/). These link are included in the text.

The following dataset was generated:

| Author(s) | Year | Dataset title | Dataset URL | Database and Identifier |
|---|---|---|---|---|
| Demir M, Kadakia N, Anderson HD, Clark DA, Emonet T | 2020 | Data from: Walking *Drosophila* navigate complex plumes using stochastic decisions biased by the timing of odor encounters | https://doi.org/10.5061/dryad.4j0zpc87z | Dryad Digital Repository, 10.5061/dryad.4j0zpc87z |

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
