## [Decision Letter]

**Acceptance summary:**

In this manuscript, the authors present the first dataset that simultaneously monitors behavior and odor in real time for freely walking flies navigating toward an intermittent odor. Flies were visualised as they navigated toward the source of a turbulent plume of smoke, that the flies were naturally attracted to. A quantitative statistical analysis of behavior in relation to odor unveiled novel algorithms underlying navigation toward intermittent odor cues.

These results pave the way for further research on the neural and computational basis of olfactory navigation strategies in the fly and introduce an attractive odor that can be imaged simultaneously with behavior, with impact for a broad swathe of the scientific community.

**Decision letter after peer review:**

Thank you for submitting your article "Walking *Drosophila* navigate complex plumes using stochastic decisions biased by the timing of odor encounters" for consideration by *eLife*. Your article has been reviewed by three peer reviewers, and the evaluation has been overseen by a Reviewing Editor and Ronald Calabrese as the Senior Editor. The following individuals involved in review of your submission have agreed to reveal their identity: Venkatesh N Murthy (Reviewer #2); Antonio Celani (Reviewer #3).

The reviewers have discussed the reviews with one another and the Reviewing Editor has drafted this decision to help you prepare a revised submission.

Summary:

In this manuscript, the authors present the first dataset that simultaneously monitors behavior and odor in real time for freely walking flies navigating toward an intermittent odor. Flies were visualised as they navigated toward the source of a turbulent plume of smoke, that the flies were naturally attracted to. Intermittency was quantified by visualisation of the smoke in real time while following the animals. A quantitative statistical analysis of behavior in relation to odor unveiled novel algorithms underlying navigation toward intermittent odor cues. Turning and stopping was a semi-random process whereas anemotactic responses depended on odor encounter. The authors tested mutants to ensure that behavior was driven by olfaction.

These results pave the way for further research on the neural and the computational basis of olfactory navigation strategies in the fly and introduce an attractive odor that can be imaged simultaneously with behavior, with impact for a broad swathe of the scientific community.

Essential revisions:

A) Because smoke is a complex signal, the behavior it elicits may be caused by a combination of effects caused by the single compounds contained in the plume, including potential repellents (e.g. CO and toluene are toxic at high concentration). Hence aspects of this behavior may be odor-dependent, and not be generally applicable to other odors. The authors should make this caveat clear to the audience, and we expect this will stimulate further work.

B) The distribution for the simulated fly (Figure 7C) is quite different from the one of true flies. One notable feature is that the simulated agent's distribution is less "peaky" along the y axis (compare with Figure 1K). True flies, it would seem, remain closer to the central axis than simulated ones. Perhaps a couple of slight changes in the model could influence/improve the fit:

• From Figure 4D, it seems like turn angle distribution could be tetra-modal, with an encounter independent part centered on 50 and an encounter-dependent part centered on 25. Could a model with this added level of granularity improve the fits to behavior? It would make odor driven changes in direction sharper and potentially improve tracking.

• The generating function for P(upwind|turn) is not very satisfactory. It is a linear fit offset by 0.5 and truncated at 1. Why did the authors not use a standard choice model with a bias term and inverse temperature controlling the slope? This could potentially handle better the extreme of the encounter frequency axis which are those that have the worst fit with the current model (Figure 4G).

C) We expect the combined behavior-odor dataset to prove extremely valuable for further research, is it possible to make it publicly available?

---

## [Author Response]

Essential revisions:A) Because smoke is a complex signal, the behavior it elicits may be caused by a combination of effects caused by the single compounds contained in the plume, including potential repellents (e.g. CO and toluene are toxic at high concentration). Hence aspects of this behavior may be odor-dependent, and not be generally applicable to other odors. The authors should make this caveat clear to the audience, and we expect this will stimulate further work.

To address this point, we added the following paragraph at the end of the Discussion: “Our visualizable signal is conventional smoke, a complex odor consisting of various aromatic compounds (Figure 1—figure supplement 1A). […] Moreover, *Orco^-/-^* flies that lack major olfactory input, but are intact in CO_2_ sensing, showed mild aversion around the center of the straight smoke plume, illustrating how different components contribute to the perception of the odor mixture.”

B) The distribution for the simulated fly (Figure 7C) is quite different from the one of true flies. One notable feature is that the simulated agent's distribution is less "peaky" along the y axis (compare with Figure 1K). True flies, it would seem, remain closer to the central axis than simulated ones. Perhaps a couple of slight changes in the model could influence/improve the fit:• From Figure 4D, it seems like turn angle distribution could be tetra-modal, with an encounter independent part centered on 50 and an encounter-dependent part centered on 25. Could a model with this added level of granularity improve the fits to behavior? It would make odor driven changes in direction sharper and potentially improve tracking.• The generating function for P(upwind|turn) is not very satisfactory. It is a linear fit offset by 0.5 and truncated at 1. Why did the authors not use a standard choice model with a bias term and inverse temperature controlling the slope? This could potentially handle better the extreme of the encounter frequency axis which are those that have the worst fit with the current model (Figure 4G).

Regarding the first suggestion, to test the multimodality of the turning distribution (Figure 4D), we used a slight variation of our turning model, in which we set the turn angle at 25^o^ for encounter frequencies of 4Hz or higher and to 40^o^ for encounter frequencies of 0 Hz, linearly interpolating between these. We did not find any significant changes in the fit of the model to the turning bias (Figure 4G). We therefore retained the turn angle distributions used in the original version (Gaussians with a mean of 30^o^ and deviation of 10^o^).

Regarding the second suggestion, we initially opted for a truncated linear model since it requires only a single parameter (assuming that the bias is 50% at an encounter frequency of 0 Hz), while a sigmoidal-type model would require 2 parameters (half-max and spread). However, it is true that the latter can similarly be reduced to a single-parameter model if we assume that the half-max is fixed to 0 Hz. This assumption is consistent with our observation that there is no turning bias for low encounter frequencies (Figure 4D). Given this assumption, we used a sigmoidal model and estimated the parameters as before, finding a spread (i.e. inverse temperature) of 0.242 Hz^-1^. Comfortingly, the parameters are fairly well localized, indicating robustness as in our linear model (new Figure 4F), and the response functions for the two models are quite similar in the 0-3Hz regime, suggesting that the sigmoidal model is largely consistent with the linear model. Further, the fits are indeed improved at the extrema, as predicted by the reviewer (new Figure 4G). Since this sigmoidal relationship has a natural asymptotic behavior at high frequency, we have replaced our truncated linear model with the sigmoidal model, updating the main text, figure, legend, and Materials and methods accordingly.

We have incorporated this updated turning model in our agent-based simulation (Figure 7C). We note that in comparing the pdfs of real and virtual flies (Figures 1K and 7C, respectively), the front end of the plume, near the lateral perturbing jets that create intermittency, the plume is not composed of odor packets but is instead a meandering ribbon. We have observed that flies zigzag around this ribbon-like region much like they do in the static ribbon (Figure 1—figure supplement 1B). Furthermore, upon reaching the source region near the odor inlet, flies aggregate around and revisit this region. Our simulations do not include these boundary-like effects, as this would require further assumptions on their behavior when hitting the arena boundaries. As such, we have removed the front 20 mm of the arena from the marginals. In combination, both the updated turning model and this slight modification in our definition of the marginals brings the virtual and real flies into more agreement (updated Figure 7C). We have noted this in the caption of Figure 7. Finally, we note that one difference between the real and virtual flies is that the former are initialized at random positions throughout the arena (once being aspirated into the device), while the latter are initialized uniformly at the back end of the arena (Materials and methods) to make the simulations more straightforward to interpret. This produces further differences between the pdfs for real and virtual flies.

C) We expect the combined behavior-odor dataset to prove extremely valuable for further research, is it possible to make it publicly available?

We have the uploaded the combined behavior-odor dataset on Dryad for anyone to download. A link is included in the paper.